# Monitoring training and recovery responses with heart rate measures during standardized warm-up in elite badminton players

**Christoph Schneider**[1]*, **Thimo Wiewelhove**[1], **Shaun J. McLaren**[2], **Lucas Röleke**[3], **Hannes Käsbauer**[4], **Anne Hecksteden**[5], **Michael Kellmann**[6,7], **Mark Pfeiffer**[8], **Alexander Ferrauti**[1]

1 Department of Training & Exercise Science, Faculty of Sport Science, Ruhr University Bochum, Bochum, Germany, 2 Department of Sport and Exercise Sciences, Durham University, Durham, United Kingdom, 3 Department of Medicine, Training and Health, Institute of Sport Science and Motology, Philipps-University Marburg, Marburg, Germany, 4 German Badminton Association, Saarbrücken, Germany, 5 Institute of Sports and Preventive Medicine, Saarland University, Saarbrücken, Germany, 6 Unit of Sport Psychology, Faculty of Sport Science, Ruhr University Bochum, Bochum, Germany, 7 School of Human Movement and Nutrition Sciences, The University of Queensland, St. Lucia, Australia, 8 Department of Theory and Practice of Sports, Institute of Sport Science, Johannes Gutenberg University Mainz, Mainz, Germany

* christoph.schneider-a5c@ruhr-uni-bochum.de

**Data Availability Statement:** The original dataset is provided as Supporting Information file. In addition, a full reproducibility documentation

## Abstract

### Purpose

To investigate short-term training and recovery-related effects on heart rate during a standardized submaximal running test.

### Methods

Ten elite badminton players (7 females and 3 males) were monitored during a 12-week training period in preparation for the World Championships. Exercise heart rate (HRex) and perceived exertion were measured in response to a 5-min submaximal shuttle-run test during the morning session warm-up. This test was repeatedly performed on Mondays after 1–2 days of pronounced recovery ('recovered' state; reference condition) and on Fridays following 4 consecutive days of training ('strained' state). In addition, the serum concentration of creatine kinase and urea, perceived recovery–stress states, and jump performance were assessed before warm-up.

### Results

Creatine kinase increased in the strained compared to the recovered state and the perceived recovery–stress ratings decreased and increased, respectively (range of average effects sizes: |d| = 0.93–2.90). The overall HRex was 173 bpm and the observed within-player variability (i.e., standard deviation as a coefficient of variation [CV]) was 1.3% (90% confidence interval: 1.2% to 1.5%). A linear reduction of -1.4% (-3.0% to 0.3%) was observed in HRex over the 12-week observational period. HRex was -1.5% lower (-2.2% to

including all data and analysis files, as well as analysis code are available via the Open Science Framework at https://doi.org/10.17605/osf.io/up4ht. The audio file for the submaximal shuttle-run test is available at https://doi.org/10.17605/osf.io/znkge.

**Funding:** The current study was funded by the German Federal Institute of Sport Science (http://www.bisp.de). The research was realized in the project REGman–Optimization of Training and Competition: Management of Regeneration in Elite Sports (IIA1-081901/17-20). Funding were received by MK MP AF. We further acknowledge support by the DFG Open Access Publication Funds of the Ruhr-Universität Bochum. The funders had no role in study design, data collection and analysis, decision to publish, or preparation of the manuscript.

**Competing interests:** The authors have declared that no competing interests exist.

-0.9%) in the strained compared to the recovered state, and the standard deviation (as a CV) representing interindividual variability in this response was 0.7% (-0.6% to 1.2%).

## Conclusions

Our findings suggest that HRex measured during a standardized warm-up can be sensitive to short-term accumulation of training load, with HRex decreasing on average in response to consecutive days of training within repeated preparatory weekly microcycles. From a practical perspective, it seems advisable to determine intra-individual recovery–strain responses by repeated testing, as HRex responses may vary substantially between and within players.

## Introduction

Today's elite athletes are often faced with a busy training and competition schedule. Coaches often seek supportive tools to make more efficient use of training time while maximizing adaptation and performance improvements. Systematic and comprehensive monitoring of athletes' short- and long-term training responses (i.e., recovery status and fitness, respectively) may help manage training load and recovery during intensive training periods. Heart rate (HR) monitoring has been well established as an inexpensive, time-efficient, and non-invasive tool in research and practice. Within a comprehensive athlete monitoring system, HR measures can represent valuable information on athletes' training responses, as they have been proposed to indicate the status of the cardiac autonomic nervous system and cardiovascular fitness [1–7]. However, the various HR measures differ in their physiological determinants and their time course of adaptation, and they display different associations to changes in fitness, fatigue, and performance [1, 8]. For example, exercise HR (HRex) is often suggested to be associated with (positive) aerobic training adaptation, while resting HR measures might also be sensitive to fatigue [1, 6].

Exercise HR recordings during standardized submaximal exercise bouts are especially attractive, as they can be performed simultaneously with an entire squad during warm-up [6]. Current monitoring technologies provide practitioners with live online feedback on players' exercise responses (i.e., relative intensity and internal load) and generally allow for easy data processing after data collection. Nevertheless, the mechanisms of HR responses are not yet fully understood, and the interpretation of changes in HR measures is not always straightforward [1, 3, 6]. For example, decreased HRex is typically associated with increased aerobic fitness and performance [1] but may also be observed during overreaching [8]. It has therefore been suggested that HR measures be interpreted considering the training context and in combination with additional subjective markers and non-invasive performance measures [1, 3, 6].

In complex sports like team and racket sports, it can be difficult to evaluate the isolated effects of certain training load characteristics, such as training volume and intensity. Different training content and exercise modalities overlap considerably within training sessions and days, and traditional measures of external (e.g., distance–time-based) and internal load (e.g., HR, blood lactate, and ratings of perceived exertion [RPE]) do not always reflect the specific physical demands. As suggested [6], we believe that contextualizing HR measures in sports with complex training structures should focus especially on the time course of training to be able to further differentiate between short-term and long-term training responses. This may ultimately help to understand so-called counterintuitive training responses [9].

In this study, a previously proposed approach [10, 11] was used to compare monitoring markers in elite badminton players at two contrasting time points during repeated weekly microcycles (recovered versus strained state) as part of a preparation period for the World Championships. We took capillary blood samples and collected self-reported recovery–stress measures prior to practice and incorporated simple physical tests in the general warm-up routine. The aim of this study was to evaluate the sensitivity of HRex during a standardized submaximal shuttle-run test in response to habitual short-term changes in training load within repeated training weeks. Our objective was to determine if HRex can differentiate between different states on the fatigue–recovery continuum (i.e., recovered versus strained) and whether potential responses can be consistently observed at the individual level.

## Materials and methods

### Participants

Twelve elite badminton players, training at the same National Training Center of the German Badminton Association, volunteered to participate in the study. All players provided written informed consent and could withdraw without penalty at any time. Ten players (7 females and 3 males, age 23 ± 4 years) were included for the analysis, providing at least two data points for both recovery states (see the section on the *study design*). Nine players were members of the German national squad, and one player received national squad status after the investigation period. The study was approved by a local Human Research Ethics Committee (Ärztekammer des Saarlandes, approval no. 228/13 and amendments) and conducted in accordance with the guidelines of the Declaration of Helsinki.

### Study design

Our study comprised a 12-week observational period during which players prepared for the World Championships. Players were tested on Mondays and Fridays at the beginning of the morning practice sessions (approximately 7:45–8:30 AM, temperature range 22.7–27.3˚C) on a total of 18 testing days (9 Mondays and 9 Fridays, Fig 1). Testing days were chosen to represent different states on the recovery–fatigue continuum, which are typically observed during repeated habitual microcycles. Training was planned by the national coaches without research team interference. According to the coaches, this preparatory period was categorized as an *intensified* training period. Training plans were provided, and the players documented their training in the best possible way. Two representative weekly training plans are detailed in Table 1, and an exemplary 12-week time course of the training volume distribution is illustrated in Fig 2. Monday values were categorized as 'recovered' state (Recovery) after 1–2 days of pronounced recovery, whereas Friday values represented a 'strained' state (Strain) following 4 consecutive days of training with up to 2 sessions per day (Table 1, Fig 2). This study design has been shown to display different levels of muscle recovery using serum concentrations of creatine kinase (CK) and urea in endurance athletes [10] and badminton players [11].

Upon arrival, players were provided with HR chest straps. Capillary blood samples were collected for determining serum concentrations of CK and urea, and players were asked to rate their perceived recovery and stress using the Short Recovery and Stress Scale (SRSS) [12, 13]. Following individual physical preparation, jump performance was assessed using countermovement jump (CMJ) and multiple rebound jump (MRJ) [14] tests. Finally, the players performed standardized submaximal shuttle-runs for approximately 5 min to start their on-court warm-ups (Fig 1).

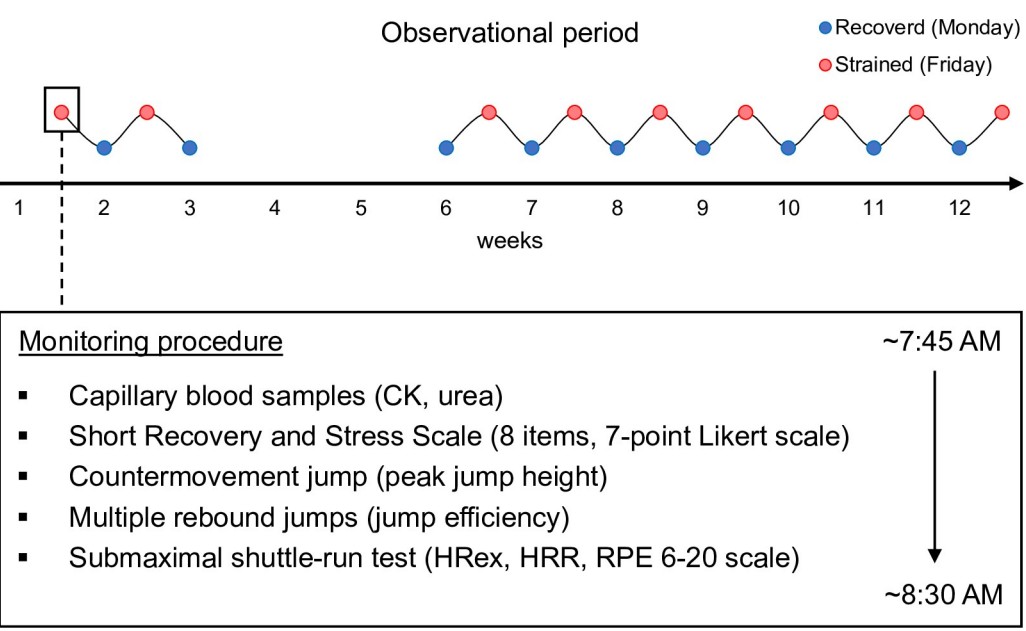

**Fig 1. Study design.** CK: creatine kinase; HRex: exercise heart rate; HRR: heart rate recovery; RPE: rating of perceived exertion.

## Procedures

**Blood markers.** Creatine kinase and urea were determined from capillary blood samples taken from a hyperemic earlobe using a 200 μL capillary blood collection system (KABE Labortechnik GmbH, Nümbrecht-Elsenroth, Germany). Samples were positioned upright to clot at room temperature for approximately 10 min, centrifuged at 6000 rpm for 10 min (Sprout, Biozym Scientific GmbH, Hessisch Oldendorf, Germany) and analyzed by the COBAS INTEGRA 400 plus (Roche Diagnostics, Basel, Switzerland).

**Short recovery and stress scale.** Perceived recovery–stress states were assessed using the SRSS, which consists of a *Short Recovery Scale* and a *Short Stress Scale* with 4 items each and responses ranging from 0 (does not apply at all) to 6 (fully applies) [12, 13]. In this study, only the physical and overall recovery and stress items were analyzed, i.e., *Physical Performance Capability* (PPC), *Overall Recovery* (OR), *Muscular Stress* (MS), and *Overall Stress* (OS). Each item is provided with 4 descriptive adjectives: PPC: *strong, physically capable, energetic, full of power*; OR: *recovered, rested, muscle relaxation, physically relaxed*; MS: *muscle exhaustion, muscle fatigue, muscle soreness, muscle stiffness*; OS: *tired, worn-out, overloaded, physically exhausted*. The internal consistency for the *Short Recovery Scale* and the *Short Stress Scale* were deemed acceptable (Cronbach's Alpha 0.72 and 0.75, respectively [13]) and previous research indicated the SRSS's sensitivity to training overload [15]. Printed versions of the SRSS that were initially used were replaced in the course of the study by an online athlete monitoring system (*REGmon—Regeneration management through athlete monitoring*) developed by the project team.

**Jump tests.** Players performed three maximal CMJs with hands on hips and self-selected rest between jumps [14]. Peak jump height, calculated from flight time, was used for analysis. Subsequently, players were instructed to perform repeated rebound jumps (i.e., MRJ) with hands on hips for approximately 15 s, focusing on maximal jump height while keeping ground contact times as short as possible. The jump efficiency coefficient (EC) was calculated by:

**Table 1. Two exemplary weekly training plans for player J.**

*Study Week 8*

| Monday | | Tuesday | | Wednesday | | Thursday | | Friday | | Saturday/Sunday | |
|---|---|---|---|---|---|---|---|---|---|---|---|
| **Session 1—AM** | | **Session 1—AM** | | **Session 1—AM** | | **Session 1—AM** | | **Session 1—AM** | | **Session 1—PM** | |
| Duration: 165 min | | Duration: 165 min | | Duration: 135 min | | Duration: 165 min | | Duration: 165 min | | Duration: 50 min | |
| Intensity: moderate | | Intensity: mod.—hard | | Intensity: low | | Intensity: hard | | Intensity: hard | | Intensity: low | |
| *15 min | *Monitoring* | | | | | | | *15 min | *Monitoring* | | |
| 30 min | Warm-up | 30 min | Warm-up | 30 min | Warm-up | 30 min | Warm-up | 30 min | Warm-up | 50 min | Endurance |
| 45 min | Strength | 45 min | Strength | 90 min | Badminton | 45 min | Strength | 45 min | Strength | | |
| 15 min | Speed | 15 min | Speed | 15 min | Cool-down | 15 min | Speed | 15 min | Speed | | |
| 60 min | Individual | 60 min | Badminton | | | 60 min | Badminton | 60 min | Badminton | | |
| 15 min | Cool-down | 15 min | Cool-down | | | 15 min | Cool-down | 15 min | Cool-down | | |
| **Session 2—PM** | | **Session 2—PM** | | | | **Session 2—PM** | | | | | |
| Duration: 120 min | | Duration: 50min | | | | Duration: 145 min | | | | | |
| Intensity: moderate | | Intensity: low | | | | Intensity: hard | | | | | |
| 15 min | Warm-up | 50 min | Endurance | | | 30 min | Warm-up | | | | |
| 90 min | Badminton | | | | | 70 min | Endurance | | | | |
| 15 min | Cool-down | | | | | 30 min | Badminton | | | | |
| | | | | | | 15 min | Cool-down | | | | |

*Study Week 9*

| Monday | | Tuesday | | Wednesday | | Thursday | | Friday | | Saturday/Sunday | |
|---|---|---|---|---|---|---|---|---|---|---|---|
| **Session 1—AM** | | **Session 1—AM** | | **Session 1—AM** | | **Session 1—AM** | | **Session 1—AM** | | **Session 1—PM** | |
| Duration: 165 min | | Duration: 165 min | | Duration: 170 min | | Duration: 165 min | | Duration: 165 min | | Duration: 50 min | |
| Intensity: moderate | | Intensity: mod.—hard | | Intensity: low | | Intensity: hard | | Intensity: hard | | Intensity: low | |
| *15 min | *Monitoring* | | | | | | | *15 min | *Monitoring* | | |
| 30 min | Warm-up | 30 min | Warm-up | 30 min | Warm-up | 30 min | Warm-up | 30 min | Warm-up | 50 min | Endurance |
| 45 min | Strength | 45 min | Strength | 90 min | Badminton | 45 min | Strength | 45 min | Strength | | |
| 15 min | Speed | 15 min | Speed | 50 min | Endurance | 15 min | Speed | 15 min | Speed | | |
| 60 min | Individual | 60 min | Badminton | | | 60 min | Individual | 60 min | Badminton | | |
| 15 min | Cool-down | 15 min | Cool-down | | | 15 min | Cool-down | 15 min | Cool-down | | |
| **Session 2—PM** | | **Session 2—PM** | | | | **Session 2—PM** | | **Session 2—PM** | | | |
| Duration: 120 min | | Duration: 135 min | | | | Duration: 145 min | | Duration: 50 min | | | |
| Intensity: moderate | | Intensity: hard | | | | Intensity: hard | | Intensity: low | | | |
| 15 min | Warm-up | 30 min | Warm-up | | | 30 min | Warm-up | 50 min | Endurance | | |
| 90 min | Badminton | 90 min | Badminton | | | 70 min | Endurance | | | | |
| 15 min | Cool-down | 15 min | Cool-down | | | 30 min | Badminton | | | | |
| | | | | | | 15 min | Cool-down | | | | |

*: Monitoring procedures started approx. 15 min prior to practice and the first 15 min were not considered training time; Badminton: individual technical training, group training with technical-tactical focus & footwork drills; Cool-down: active recovery, foam rolling & stretching; Endurance: intensive on-court drills & extensive off-court exercise; Individual: individualized strength & weaknesses; Match: competitive match play & games; Speed: on-court drills & off-court exercises for speed & speed-endurance; Strength: whole-body strength training; Warm-up: individual & coach-guided.

EC = flight time$^2$ / ground contact time / 1000. Based on the EC, the best five jumps were selected, and the mean EC was used for analysis [14]. Jump tests were performed on a contact platform (Haynl-Elektronik GmbH, Schönebeck, Germany). Test-retest reliability was assessed in our laboratory using published spreadsheets [16] (unpublished results: peak CMJ (cm), n = 38, intraclass correlation coefficient [ICC (3,1)] = 0.85, standard error of measurement (SEM) = 1.88, coefficient of variation (CV) = 4.6%; MRJ (EC), n = 38, ICC (3,1) = 0.87, SEM = 0.10, CV = 7.3%).

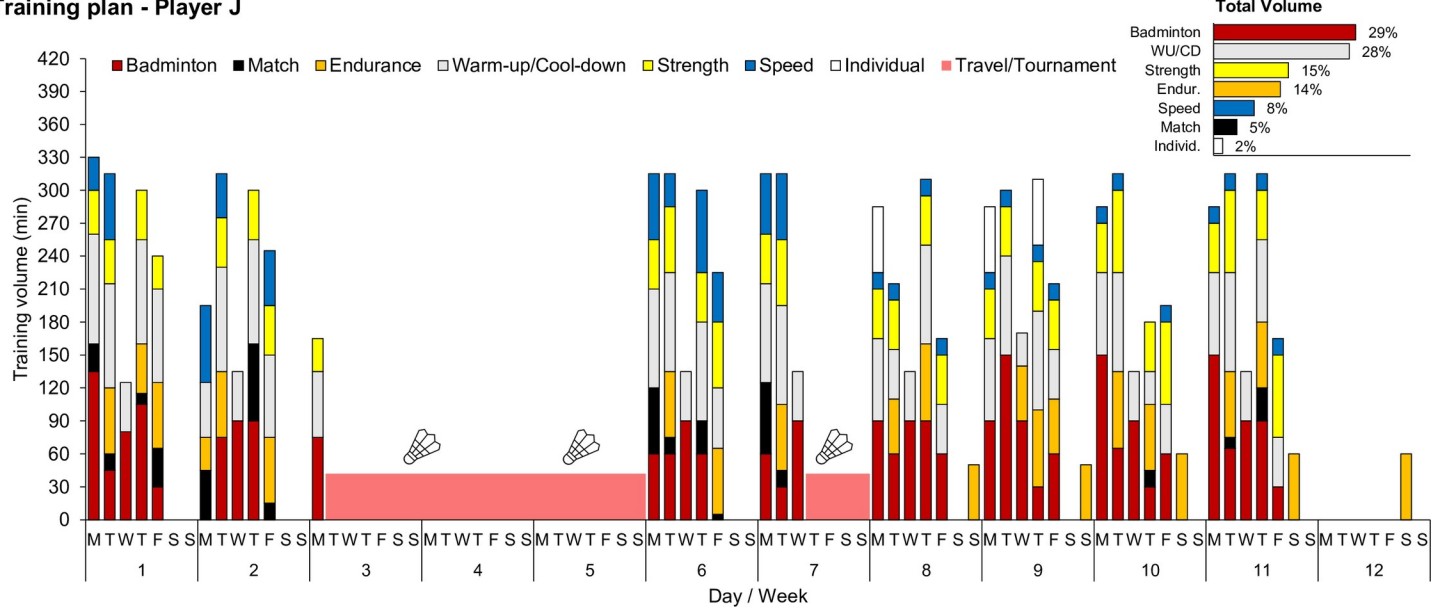

**Fig 2. Time course of the training volume distribution over the 12-week study period in player J.**

**Submaximal shuttle-run test.** In the absence of clear test recommendations, based on previous research experience [6], and based on pragmatic considerations by the coaches regarding the test nature, we used a tailor-made shuttle-run protocol to assess HRex and RPE (6–20 scale) as part of a standardized on-court warm-up routine (Fig 3). In contrast to established intermittent tests (e.g., sub-maximal Yo-Yo tests [17, 18]), coaches and we preferred a continuous exercise test to obtain more stable heart rate data while using short-distance shuttles for movement specificity. We initially sought to assess HR recovery (Fig 1), but due to

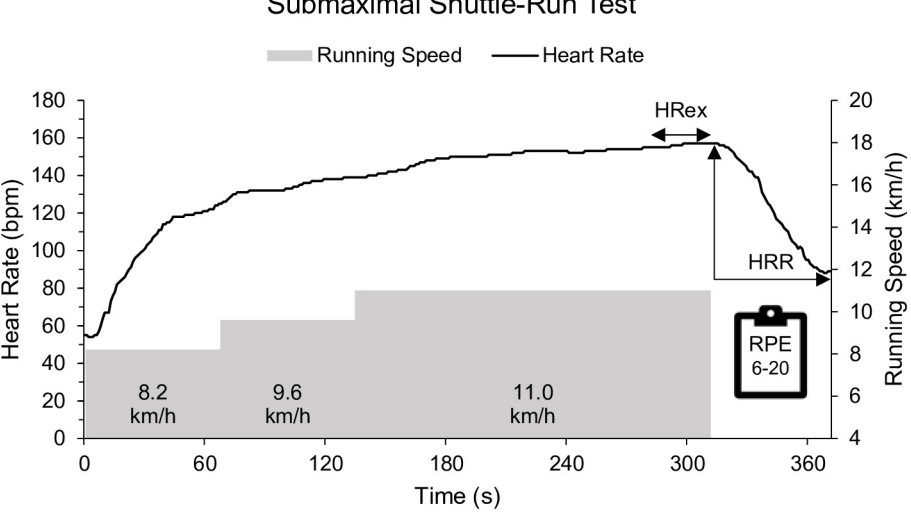

**Fig 3. Submaximal shuttle-run test.** Players run 12.8 m shuttles for approximately 1, 1, and 3 min at 8.2 km/h, 9.6 km/h, and 11.0 km/h, respectively, followed by 1 min of standing recovery. HRex: average heart rate during the last 30 s of exercise; HRR: heart rate recovery during the 1-min recovery period; RPE: rating of perceived exertion (6–20 scale) at exercise cessation.

athletes talking and moving during the one-minute recovery time, the data quality was insufficient for analysis (only 57/95 available recordings satisfied the inclusion criteria). The submaximal shuttle-run test consisted of 12.8 m shuttle-runs for approximately 5 min, followed by 60 s of passive recovery, performed across two official badminton playing fields at the National Training Center (YONEX® court mats, certified BWF standard—Grade 1). The test was based on a modified protocol originally developed for HR monitoring in semi-professional basketball players [cf. 6]. The average running speed was set at 8.2 km/h, 9.6 km/h, and 11.0 km/h for 67.2 s, 67.2 s, and 176.4 s, respectively. This corresponds to 12, 14, and 42 shuttles per stage in 5.6 s, 4.8 s, and 4.2 s per shuttle, respectively. Average running speed was calculated as the total distance per stage divided by stage duration. There was no speed adjustment for changes of direction. The beginning of each stage, the end of the final stage, and the recovery period were indicated by respective audio signals. The submaximal shuttle-run test audio file is available online at https://doi.org/10.17605/osf.io/znkge [19].

Heart rate was recorded with the acentas™ Team HR monitoring system (Heart Rate Monitoring System for USBRTX3, version 2.09, acentas GmbH, Hörgertshausen, Germany) and stored and processed with 1 HR value per second. Heart rate recordings were visually inspected for data quality. Subsequently, HRex was calculated as the mean HR during the last 30 s of exercise. Heart rate data were analyzed in original units (beats per minute [bpm]), as validated peak HR data were neither available nor verifiable during the study period. The Borg 6–20 RPE scale was printed in A4 format and placed at multiple points next to the finish line. The players were asked to look at the scale and rate their perceived exertion at exercise cessation. Immediately after recovery, players marked their RPE score on a personal printout to avoid verbal interference between the athletes.

## Statistical analysis

Descriptive data are presented as the mean ± standard deviation (SD) unless otherwise specified.

The first part of our analysis was performed in Microsoft Excel 2016 (Microsoft Office 365, Version 2004, Microsoft Corp., Redmond, WA, USA) and the free statistics packages JASP (Version 0.12.0, Amsterdam, Netherlands [20]) and jamovi (Version 1.2.16 [21]). We descriptively compared individual mean HRex between recovered and strained states before comparing changes from Monday to Friday (ΔStrain) and changes from Friday to Monday (ΔRecovery). These change scores represent sets of paired samples, where a player registered consecutive testing data within a given training microcycle. Mean differences are presented in raw units. Creatine kinase und urea were analyzed using log-transformed data (i.e., natural logarithms), and the results were back-transformed for presentation. Standardized differences ($d$) were also calculated from the pooled within-player SD for recovered states [18 p.289].

In the second part of our analysis, we determined the overall mean difference in HRex between recovered and strained states and quantified interindividual differences in this response. To account for the hierarchical data structure and control for the overall change (trend) in HRex over the 12-week observation period, data were analyzed using a within-player linear mixed effects model. Models were run using the MIXED procedure in SAS® software (University Edition, SAS Institute Inc., Cary, NC, USA) via Restricted Maximum Likelihood and with the Kenward-Roger denominator degrees of freedom method [22]. Since the minimum practically important change in HRex is said to be approximately 1% over 'moderately-long' training periods [1], analysis was performed on the log-transformed data so that outputs could be expressed in percentage units.

The model fixed effects were state (categorical factor: recovered or strained) and training week (continuous covariate: mean-centered and re-scaled, ranging from -0.5 to 0.5 to account for the overall linearized change in HRex across the 12-week observation period). We included random effects for Player ID (intercept) and Player ID × Week (slope) to allow for individual differences in HRex and the linearized change over the 12-week period. A random effect was also added for Player ID × State to determine interindividual variability in the mean (fixed) state effect. We used SAS® code supplied by Goltz et al. [23], which is a modified version of that proposed by Senn et al. [24], to include a covariate "dummy" variable (XVare) designed to derive the true individual response variance [19]. All random effects were specified with a variance components covariance structure and expressed as CVs (i.e., SDs in percentage). The model appropriateness was verified by examining the plots of the studentized residual and predicted values. All fixed and random effects are presented with 90% confidence intervals (CIs).

We applied a minimum effect test (MET) [25] to provide a practical, probabilistic interpretation of the difference in HRex between the recovered and strained states. The MET aims to combine the strength of drawing inferences from the data in relation to meaningful effect sizes with a formal statistical foundation grounded in frequentist approaches to inferences [26]. The MET was performed as part of the MIXED procedure in SAS® Software, using –1% as the threshold for practical importance. Due to the exploratory nature of our analysis, probability values for the one-sided tests ($P_{MET}$) were presented as continuous estimates.

Finally, we estimated the proportion of true responders in HRex using a recently recommended approach [27–29]. This method uses the estimates of the mean short-term difference and the associated SD representing the interindividual variability to derive the proportion of interindividual differences (i.e., recovery–strain) free from (random) within-subject variability and greater than the minimum practically important difference (i.e., -1%).

All relevant data, analysis files, and code are available via the Open Science Framework at https://doi.org/10.17605/osf.io/up4ht.

## Results

Summary statistics are detailed in Table 2, and distributions of individual effect sizes are visualized in Fig 4. Creatine kinase was increased on Fridays compared to Mondays ($d$ = 1.91 ± 0.73), and the ratings of perceived physical and overall recovery and stress in the SRSS decreased (PPC, OR) and increased (MS, OS), respectively (range of average effect sizes $|d|$ = 0.93–2.90). The mean effect sizes for urea, jump tests, and RPE were $d \leq 0.5$.

A total of 95 HRex measurements, including 4–14 measurements per player, were available for analysis. Individual HRex time series for the 12-week study period are displayed in Fig 5. From these 95 observations, there were 26 and 36 pairs of individual ΔStrain and ΔRecovery change scores, respectively (i.e., changes from Monday to Friday or changes from Friday to Monday, respectively). Distributions of these change scores are displayed in Fig 6.

The overall (grand mean) HRex was 173 bmp, with between-player and within-player SDs (as CVs) of 5.8% (90% CI: 2.7% to 7.7%) and 1.3% (1.2% to 1.5%), respectively. We observed a linear reduction of -1.4% (-3.0% to 0.3%) in HRex over the 12-week study period. The interindividual variability in this trend (SD as a CV) was 2.2% (-0.6% to 3.2%). The estimated marginal means for recovered and strained HRex were 174 bpm and 171 bmp, respectively. HRex was -1.5% lower when strained compared to recovered state (-2.2% to -0.9%, $P_{MET}$ = 0.09; Fig 7). The SD representing interindividual variability in this difference was 0.7% (-0.6 to 1.2; Fig 7).

Using the mean difference of -1.5%, the interindividual response SD of 0.7%, and a minimum practically important difference of -1%, we estimated the proportion of true and substantial HRex responders to be 78%, with the remaining 22% of responses being trivial.

**Table 2. Summary of monitoring data.**

| Var. | CK** | | Urea** | | PPC | | OR | | MS | | OS | | CMJ | | EC | | HRex | | RPE | |
|---|---|---|---|---|---|---|---|---|---|---|---|---|---|---|---|---|---|---|---|---|
| Unit | (U/L) | | (mg/dL) | | (0–6) | | (0–6) | | (0–6) | | (0–6) | | (cm) | | (index) | | (bpm) | | (6–20) | |
| State | Rec | Strain | Rec | Strain | Rec | Strain | Rec | Strain | Rec | Strain | Rec | Strain | Rec | Strain | Rec | Strain | Rec | Strain | Rec | Strain |
| Tests | 52 | 57 | 52 | 57 | 49 | 47 | 49 | 47 | 49 | 45 | 49 | 45 | 51 | 56 | 51 | 53 | 48 | 47 | 51 | 51 |
| Mean | 142[#] | 265[#] | 28[#] | 30[#] | 3.8 | 3.1 | 3.8 | 2.5 | 1.8 | 3.7 | 2.1 | 3.6 | 36.0 | 36.4 | 1.41 | 1.46 | 174.0 | 171.6 | 13.9 | 14.2 |
| SD | 72* | 147* | 7* | 6* | 0.3 | 0.2 | 0.5 | 0.6 | 0.5 | 0.4 | 0.5 | 0.4 | 6.0 | 5.8 | 0.25 | 0.24 | 10.1 | 9.8 | 1.6 | 1.3 |
| $SD_{within}$ | 47* | 144* | 4* | 5* | 0.7 | 0.6 | 0.8 | 0.7 | 0.7 | 0.9 | 0.8 | 0.8 | 1.2 | 1.1 | 0.20 | 0.18 | 2.3 | 3.0 | 0.9 | 1.1 |
| Diff. | 123[#] | | 2[#] | | -0.6 ± 0.3 | | -1.4 ± 0.7 | | 1.9 ± 0.7 | | 1.5 ± 0.5 | | 0.4 ± 0.6 | | 0.05 ± 0.08 | | -2.4 ± 1.8 | | 0.3 ± 0.9 | |
| $d$ | 1.91 ± 0.73 | | 0.52 ± 0.90 | | -0.93 ± 0.48 | | -1.60 ± 0.78 | | 2.90 ± 1.04 | | 1.95 ± 0.70 | | 0.34 ± 0.54 | | 0.24 ± 0.39 | | -1.03 ± 0.79 | | 0.36 ± 0.92 | |

Individual mean data are summarized for recovered and strained state. Differences between states are presented as mean differences and standardized mean differences. CK: creatine kinase; PPC: *Physical Performance Capability* (Short Recovery and Stress Scale, SRSS); OR: *Overall Recovery* (SRSS); MS: *Muscular Stress* (SRSS); OS: *Overall Stress* (SRSS); CMJ: countermovement jump height; EC: jump efficiency coefficient (multiple rebound jumps), HRex: exercise heart rate, RPE: rating of perceived exertion.

Var.: Variable; Rec: 'recovered' state; Strain: 'strained' state; SD: standard deviation (between-player); $SD_{within}$: pooled within-player SD for recovered state; Diff.: mean difference between recovered and strained state; $d$: standardized mean difference = mean difference divided by $SD_{within}$.

Diff. and $d$ are presented as mean ± between-player SD.

[#]Mean: back-transformed means of log-transformed CK and urea (lnCK, lnUrea); [#]Diff.: difference between back-transformed grand means of lnCK and lnUrea (S1 Table for details).

*SDs for CK and urea represent the average distance between back-transformed mean–SD and mean + SD of lnCK and lnUrea (S1 Table for details).

**Mean ± SD for females only: CK Rec 120 ± 44 U/L, CK Strain 212 ± 92 U/L, Urea–Rec 26 ± 7 mg/dL, Strain 28 ± 6 mg/dL.

Individual descriptive statistics (S1 Table), player reports (S1 Appendix), variability plots for observed HRex data (S1 Fig), and HRex change scores (S2 Fig) are available as supplementary material.

## Discussion

The purpose of this observational study was to assess whether HRex during standardized sub-maximal shuttle-runs is sensitive to short-term accumulation of training load within repeated habitual training weeks and whether potential responses can be consistently observed at the individual level. The standardized effect sizes in CK and subjective recovery–stress markers characterize the appropriateness of the present study design for describing different levels of recovery and training strain during repeated preparatory training microcycles (Table 2, Fig 4). Our findings are consistent with previous studies that reported increased levels of CK 12 hours after badminton-specific training [30] as well as in response to repeated weekly microcycles in endurance athletes [10] and badminton players [11]. Furthermore, self-reported measures of recovery and stress were reduced and increased in response to training strain, respectively. The main finding of this study was that HRex was sensitive to short-term changes in training load within the current training regime. On average, and for most players, mean HRex was lower on Fridays after 4 consecutive days of training (i.e., strained state) compared to Mondays after pronounced recovery over the weekend (i.e., recovered state) (Fig 7, S1 Table). This main effect was further supported when assessing the proportion of anticipated responders, which was 78%. In most of our cases, HRex decreased from Monday to Friday and increased again from Friday to Monday (Fig 6, S2 Fig). For some players, individual responses showed quite consistent patterns (Fig 5).

Our study was based on the premise that training strain would be evident after four consecutive training days during each weekly microcycle. This premise was verified by substantial changes in CK and perceived recovery–stress ratings (Table 2, Fig 4). At the same time, the

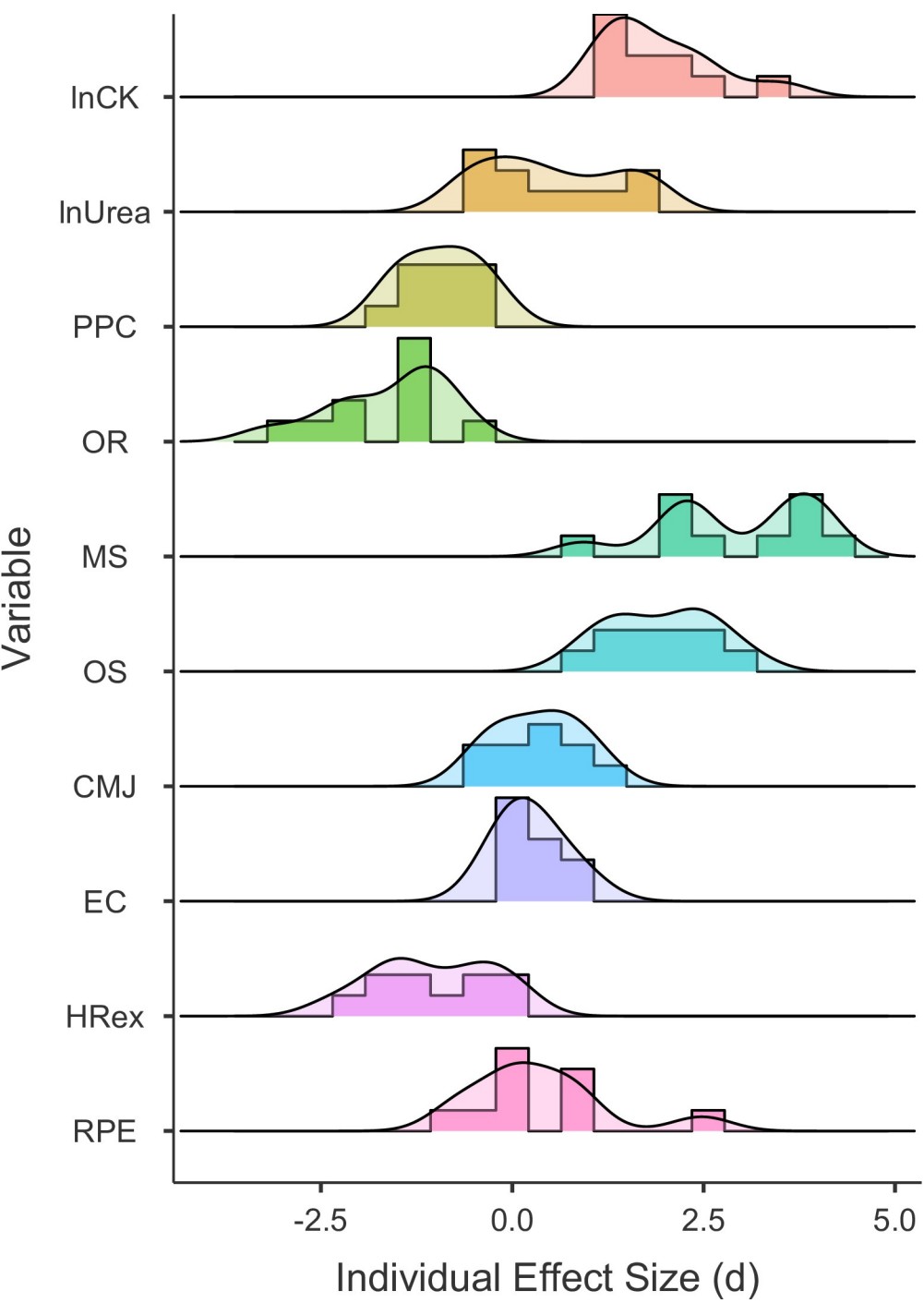

**Fig 4. Distributions of individual effect sizes between recovered and strained state for the different monitoring markers.** Plot displays individual standardized mean differences (*d*) between recovered and strained state. lnCK: natural logarithm of creatine kinase; lnUrea: natural logarithm of Urea; PPC: *Physical Performance Capability* (Short Recovery and Stress Scale, SRSS); OR: *Overall Recovery* (SRSS); MS: *Muscular Stress* (SRSS); OS: *Overall Stress* (SRSS); CMJ: countermovement jump height; EC: jump efficiency coefficient (multiple rebound jumps); HRex: exercise heart rate, RPE: rating of perceived exertion.

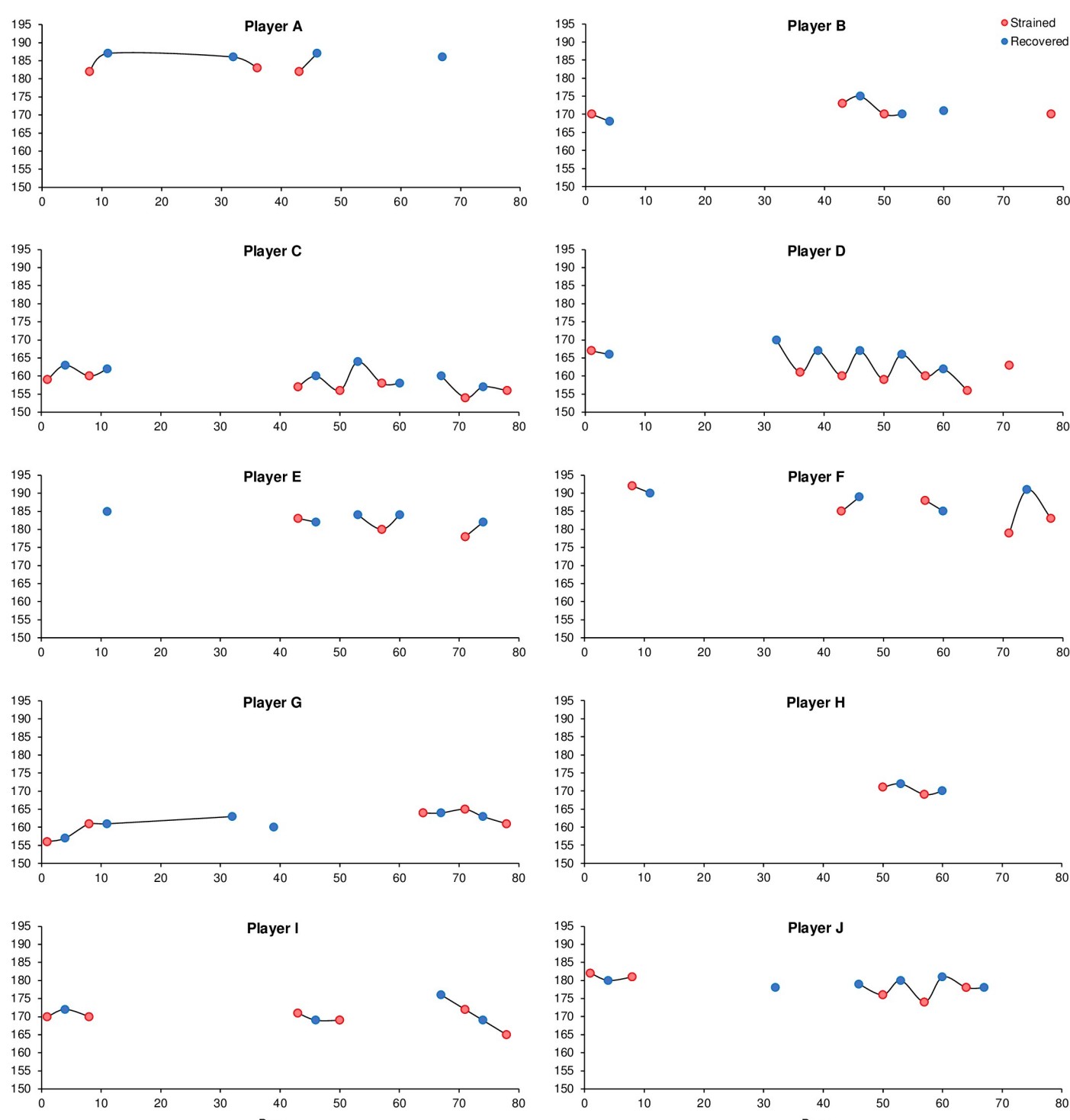

**Fig 5. Individual exercise heart rate time series.** Plot displays individual HRex time series during the 12-week study period. Lines between data points are interrupted if measurements are not available or missing. Blue dots: 'recovered' state (Mondays), red dots: 'strained' state (Fridays).

mean recovery–strain difference in HRex was -1.5% (-2.2% to -0.9%), which may be considered clear but small. It has been suggested that a reduction of approximately 1% in HRex could be defined as a minimum practically important difference when assessing positive training

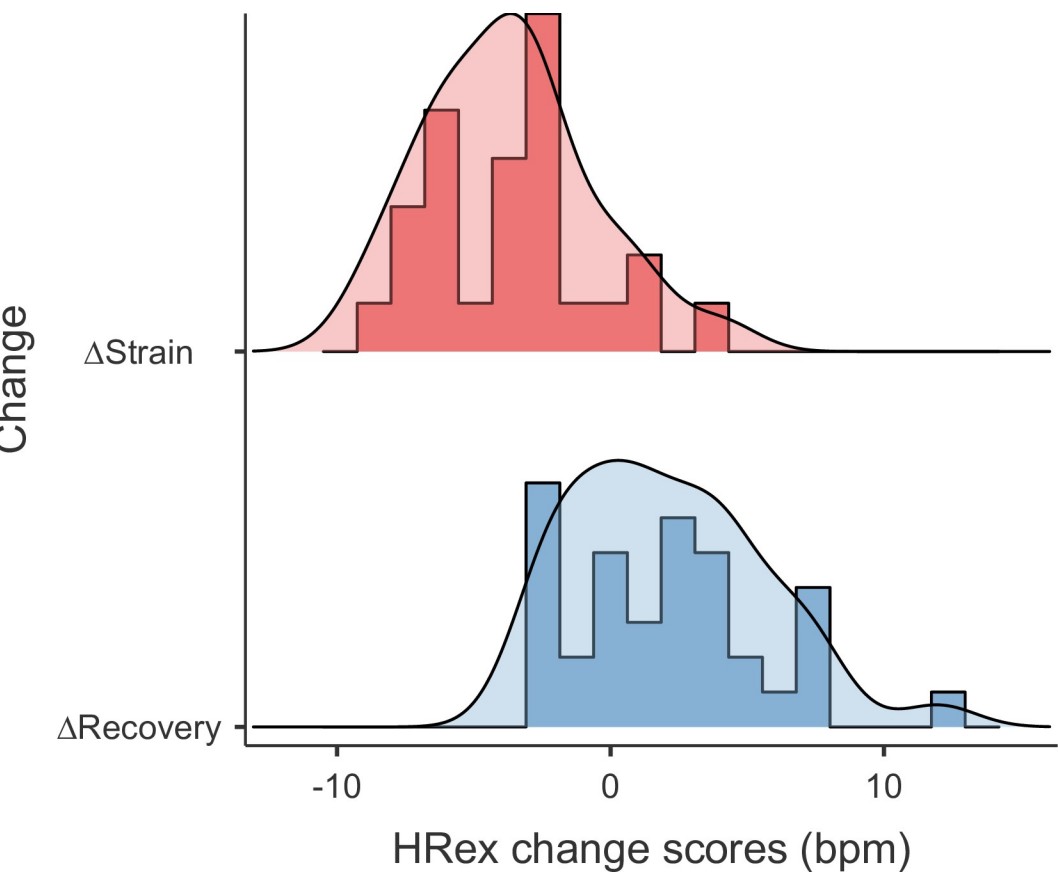

**Fig 6. Change scores in exercise heart rate (HRex).** Plot displays separate distributions of individual change scores in response to training strain and recovery. A total of 26 ΔStrain and 36 ΔRecovery change scores were available.

adaptations in 'moderately-long' training periods [1]. In our study, this could be converted to approximately 1.7 bpm when considering the grand mean HRex of 173 bpm. However, thresholds for short-term and negative training responses still need to be investigated [1]. Using this -1% threshold, the minimum effect test indicated weak compatibility [26] with a practically meaningful mean difference between recovered and strained state on the group level ($P_{MET}$ = 0.09, Fig 7). Given the interindividual variability in this short-term response of 0.7%, approximately 78% of players were estimated to have a practically meaningful true HRex response (i.e., recovery–strain difference free from (random) within-subject variability). However, it should be noted that the 90% confidence interval for the interindividual response variability was relatively wide (-0.6% to 1.2%), indicating a considerable uncertainty in this estimate. Qualitatively, average and most individual effects ($d$) ranged between small (< 0.87) and moderate (0.87–2.67) magnitudes when considering effect size thresholds, which have been proposed for analyzing A–B contrasts in single-case research designs [31 p.161] (Fig 4, S1 Table).

In the context of endurance-type overload training or overreaching, reduced HRex combined with increased measures of fatigue is a well-documented phenomenon. In their 2003 narrative review, Achten and Jeukendrup [2] concluded that overall the effects of overreaching on submaximal HRex are controversial, with individual studies reporting decreased [32–36] or unchanged HRex [37–39]. In retrospect, these so-called controversial observations may be partially due to 'non-significant' findings being interpreted as evidence for the absence of an effect. In a later systematic review, Bosquet et al. [8] estimated a small overall effect of overload

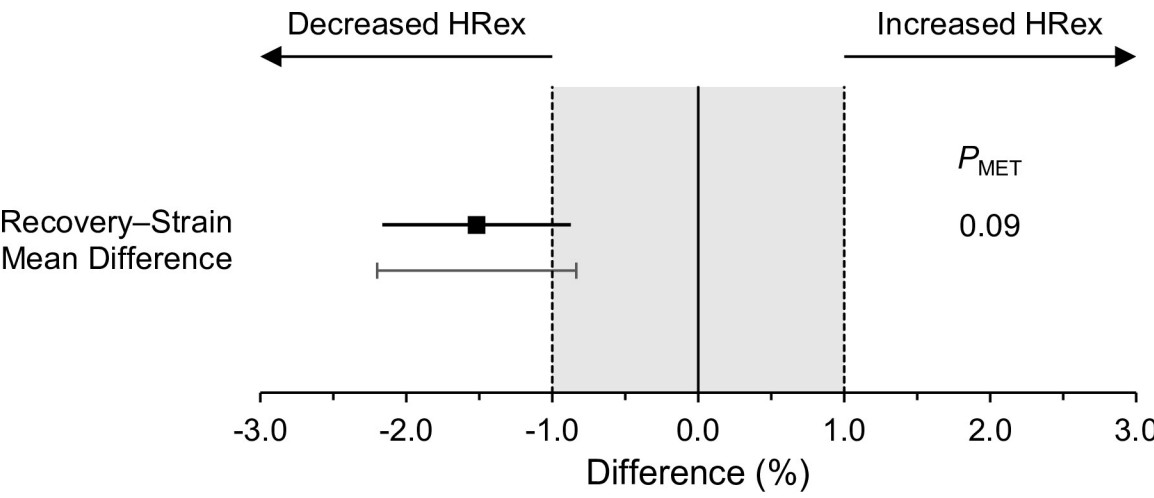

**Fig 7. Short-term effect in exercise heart rate (HRex).** Plot displays mean difference between recovered and strained state with 90% confidence interval (black square with error bars), as well as interindividual response variability around the mean effect (i.e., ± interindividual SD, gray vertical line with caps). $P_{MET}$: $P$ value for minimum effect test against a minimum practically important difference of 1% [1].

training on submaximal HRex (overall effect: -2.6 bpm; > 2 training weeks: -3.6 bpm). More recently, Le Meur et al. [40] found a substantially stronger reduction in HRex during submaximal treadmill running in triathletes during 3 weeks of ~50% increased training volume compared to triathletes training as normal. In addition, Ten Haaf et al. [41] observed a clear HRex reduction during submaximal cycling one week after an 8-day non-competitive amateur cycling event (-4.4 bpm at 80 W and -5.5 bpm at ~50% peak power output). Similarly, increased power output at a fixed percentage of maximum HR and increased perceived stress or fatigue were observed after increased weekly training loads [42] and at the end of 6-day [43] and 8-day [44] training camps. Increased power output at a fixed percentage of maximum HR can be translated to reduced HRex at fixed power output. Interestingly, substantial reductions in HRex during submaximal treadmill running were also reported 2 days after a 56-km ultramarathon [9] and 3 days after an 87-km ultramarathon [45]. Overall, a decrease in HRex appears to be well documented after endurance-type overload training and extreme endurance events. In comparison, the short-term HRex effect that we observed within repeated training microcycles (approximately -2.6 bpm) is comparable to average effects reported after an ultra-endurance event (mean differences of -3.4 bpm and -2.1 bpm at 70% and 85% peak treadmill running speed, respectively [9]) and somewhat lower than the lower range of group effects reported after overload training (mean differences of -4 to -5 bpm) [32, 33, 36, 37].

In badminton and other racket and game-based sports, sport performance is complex and multifactorial, so training must include several aspects, such as technical, tactical, physiological, and psychological components [6]. Given the complexity of training and therefore the subordinate importance of cardiovascular demands compared to endurance-type exercise, we were surprised to find a short-term decrease in HRex comparable in magnitude to changes observed shortly after an ultramarathon event [9]. This potential sensitivity of HRex to reflect naturally occurring short-term training load changes within habitual microcycles despite a variety of training contents, not solely focusing on endurance-type exercise (Table 1, Fig 2), is in line with observations by Buchheit et al. [46] and Malone et al. [47]. During 8-day and 14-day training camps in Gaelic Football [47] and Australian Rules Football [46], respectively, daily changes in HRex were strongly correlated with daily training load changes ($r \geq$ -0.8).

Overall, training loads were substantially larger than normal, and HRex decreased throughout the camps. In summary, we conclude that HRex can also decrease after several consecutive training days under normal training conditions, even if the focus is not solely on aerobic-type exercise.

Although one can only speculate about the underlying mechanisms of the observed short-term HRex changes in our study, there are several possible explanations. HR measures are often associated with cardiac autonomic nervous system activity and aerobic fitness [1, 2, 4, 48, 49]. During exercise, heart activity and therefore HR is controlled by cardiac parasympathetic and sympathetic nervous system activity [8, 49]. Hence, changes in HRex have been partially attributed to changes in autonomic nervous system status. Decreased HRex could reflect reduced sympathetic nervous system activity, reduced catecholamine tissue responsiveness, and/or changes in adrenergic receptor activity [50]. Reduced HRex has previously also been associated with increased parasympathetic activity [40, 43]. Furthermore, several studies have reported increased plasma volume after intense exercise [33, 46, 51, 52], which leads to increased stroke volume and lower HR at maintained cardiac output [33, 46]. In addition to physiological changes, several influencing factors, such as hydration status or ambient temperature, are known to alter HRex [2]. Due to the long-term observational period and the repeated within-subject contrasts design of our study, systematic differences in hydration status between Monday and Friday measurements appear unlikely. Furthermore, the ambient temperature was quite stable, with the average temperature on Fridays being ~0.5–3.1°C higher than on Mondays for the individual contrasts. In comparison to the previously reported effects of heat or cold [2], we think that the observed small temperature difference between recovered and strained states can be omitted as a potential cause for the changes in HRex. In summary, a combination of changes in plasma volume and changes in cardiac autonomic nervous system activity seems to be the most plausible explanation for the observations. Both effects are well documented in the acute and short-term phases and appear convincing under the given training conditions.

When monitoring athletes' responses in sports practice, it is essential that training and recovery effects can be observed clearly and consistently at the individual level. A within-player SD of 1.3% (1.2% to 1.5%) indicated that the intra-individual variability in HRex was smaller than a previously reported typical error (i.e., standard error of measurement) of approximately 3% [1]. In addition, the intra-individual responses of some athletes were surprisingly clear and consistent given the uncontrolled training setting of the current study, in which standardized training stimuli were not intended and gapless data were rare. For example, clear response patterns were visible by pure observation for players C, D, and, at times, player J when weeks of consecutive measurements for these players were available (Fig 5). Conversely, it could be argued that a mean HRex difference of -1.5% between the recovered and the strained state approximated to -2.6 bpm in our sample and may not appear very compelling. HRex is typically derived as an integer, and the observed mean difference is therefore only about twice the smallest observable difference at the individual level. At the same time, the interpretation of single HRex measurements or change scores is generally affected by an expected non-trivial measurement error [1]. To address the generic challenge of observed measurement error in sports practice, it seems advisable to establish intra-individual recovery–strain response profiles through repeated testing as part of a 'learning phase' [11], before decisions on training and recovery prescription are made. In summary, our findings suggest that, on average, HRex was clearly affected by naturally occurring short-term training load changes within repeated training microcycles. If decision-making at the individual level is desired, it seems advisable to incorporate repeated testing (i.e., multiple recovered versus strained measurements), as HRex responses may vary substantially between and within players.

## Limitations and strengths of the study

In the absence of quantitative training load data and an objective and accurate criterion measure, the present study was based on the premise that Monday and Friday measurements during repeated habitual training weeks display a substantial and practically relevant contrast between the recovered versus the strained time points. Although it is generally desirable to validate different recovery states against an accepted criterion measure, such as sport-specific maximum performance in the context of overreaching, there is not yet a practical alternative that can be used regularly in elite athletes' training environments. In addition, detailed training load quantification would have allowed a more complete description of the training execution and demands. Unfortunately, gapless workload monitoring data was not available for the study period. Nevertheless, the application of this repeated measures design has been shown previously to display different levels of muscle recovery in junior elite endurance athletes [10] and elite badminton players [11], which was also supported by our findings for mean CK levels. In addition, moderate to large [31] standardized mean differences in perceived recovery–stress states were present in the analyzed SRSS items underpinning the appropriateness of the study design. Although self-reported measures could potentially be manipulated by athletes and may have limitations, athlete-reported outcome measures of training response are well established [7, 53] and are considered sensitive to increased training load [54]. It must be acknowledged, however, that we had to change the survey method from a printed to a digital version of the SRSS for reasons of compliance. This was done to enable players and coaches a more frequent implementation of the SRSS in daily practice including immediate online access to data and results. Although this change in method might have influenced the SRSS ratings, we assume the intra-individual effects to be minor in our case, as players were well familiarized with the original validated print version before using the online version.

Complete datasets for several consecutive weeks were not available for all athletes. Although we tried to ensure that testing was as complete as possible, it was not possible to do so consistently given the circumstances. For example, several recordings were missing due to variations in individual training and competition schedules (i.e., Monday or Friday tests not reflecting the recovered and strained states, respectively), disease, injury, or poor HR data quality. While missing CK and urea data do not necessarily impair the development of individualized reference values, for which this study design was originally developed, missing HRex data more strongly limit the analysis and interpretation.

The novel method of determining individual responses from a within-subjects design (i.e., replicate crossover) requires randomization in the order of treatments replications at the level of the individual or at the very least some element of chance allocation involving different sequences balanced for trends [24]. Because our study was observational, conducted in applied practice, we could not balance the testing sequences order. Rather, the athletes' training schedule allowed for repeated measurements of 'control' (i.e., recovered state: reference condition) and 'intervention' (i.e., strained state, following 4 days of training) conditions. This should be considered when interpreting our findings and applying such a method to similar designs.

Finally, some athletes showed quite notable long-term changes in HRex levels, which might be related to changes in aerobic fitness and especially complicate the analysis of short-term effects in the present study. Furthermore, due to the applied nature of this study, it cannot be ruled out that observed HRex responses were confounded by strain-induced changes in movement patterns or altered player behavior (e.g., minor short cuts during changes of direction, which cannot be detected when simultaneously observing the whole training group).

However, we believe these potential limitations also make a positive contribution to the ecological validity of our findings. Elite sport practitioners are often confronted with

suboptimal circumstances that make repeated, systematic, and controlled observations challenging. In our experience, non-standardized training weeks and missing data are common. However, sophisticated analysis strategies, which adequately deal with missing data and, at the same time, can differentiate between overlapping short-term and long-term training responses, are rare. Therefore, it is likely that mean training responses that can still be consistently observed through simple descriptive analysis or even visual inspection will also be recognized by practitioners in normal training situations.

Future studies should aim to verify our observations to enhance the understanding of short-term and long-term changes in HRex. This could be done by replicating the study under more controlled (laboratory) conditions, ideally randomizing repeated (blocks of) training and control weeks [55, 56] and with a more definitive sample size. Alternatively, it may also be valuable to measure daily HRex over the course of several consecutive training weeks to assess the time course and consistency of HRex response in more detail. Nevertheless, the influence of different training characteristics (e.g., intensity, volume, and exercise mode) is still unknown, and valid quantification of concurrent different training components is very challenging, which will impede the comparability and generalizability of applied field studies. Future studies should therefore also aim to systematically and comprehensively quantify training load to allow a more direct description of the training context and to enable dose-response analyses. Finally, it might also be valuable to evaluate the effects of potential influencing factors (e.g., anthropometrics, body composition) to better understand inter-individual differences in training and recovery responses.

## Conclusions

Our findings indicate that HRex may be reduced after consecutive training days during habitual preparatory training weeks and not only in response to positive aerobic training adaptation or overload training. Despite a clear average effect, we encourage practitioners to implement repeated testing when decision-making at the individual level is desired, as HRex responses may vary substantially between and within players. Furthermore, since the HRex response is known to be influenced by many factors, practitioners should consider the potentially overlapping effects of acute and short-term training load changes, long-term training adaptations, and external confounding factors when interpreting HRex.

## Supporting information

**S1 Dataset. Original data.**
(XLSX)

**S1 Appendix. Individual player reports.** Individual monitoring data are visualized for each player during the observational period.
(PDF)

**S1 Fig. Individual exercise heart rate data (HRex).** Plot displays individual HRex for recovered (blue) and strained (red) state during the 12-week study period.
(PDF)

**S2 Fig. Change scores in exercise heart rate.** Plot displays individual change scores in HRex following recovery (ΔRecovery, blue) and following training strain (ΔStrain, red) during the 12-week study period.
(PDF)

**S1 Table. Summary of individual monitoring data.** Individual data are summarized for recovered and strained state. Group data summarizes mean individual data.
(PDF)

## Acknowledgments

We would like to thank the athletes who participated in this study as well as the coaches involved in this project. We would also like to thank all students who participated in the data collection.

## Author Contributions

**Conceptualization:** Christoph Schneider, Hannes Käsbauer, Anne Hecksteden, Alexander Ferrauti.

**Data curation:** Christoph Schneider, Shaun J. McLaren.

**Formal analysis:** Christoph Schneider, Shaun J. McLaren.

**Funding acquisition:** Michael Kellmann, Mark Pfeiffer, Alexander Ferrauti.

**Investigation:** Christoph Schneider, Thimo Wiewelhove, Lucas Röleke.

**Methodology:** Christoph Schneider, Anne Hecksteden, Alexander Ferrauti.

**Project administration:** Christoph Schneider.

**Resources:** Alexander Ferrauti.

**Supervision:** Alexander Ferrauti.

**Validation:** Christoph Schneider.

**Visualization:** Christoph Schneider.

**Writing – original draft:** Christoph Schneider.

**Writing – review & editing:** Christoph Schneider, Thimo Wiewelhove, Shaun J. McLaren, Lucas Röleke, Hannes Käsbauer, Anne Hecksteden, Michael Kellmann, Mark Pfeiffer, Alexander Ferrauti.

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
