## [Decision Letter · Decision Letter 0]

11 Aug 2020

PONE-D-20-22225

Monitoring training and recovery responses with heart rate measures during standardized warm-up in elite badminton players

PLOS ONE

Dear Dr. Schneider,

Thank you for submitting your manuscript to PLOS ONE. After careful consideration, we have decided that your manuscript does not meet our criteria for publication and must therefore be rejected. As you will see, some limits have been raised during the reviewing process. However, the major problem is that the entire text, results, figures and tables can be search open access in ResearchGate and SportRχiv, without peer-review and this migth be confonding for PLOS ONE.

I am sorry that we cannot be more positive on this occasion, but hope that you appreciate the reasons for this decision.

Yours sincerely,

Laurent Mourot

Academic Editor

PLOS ONE

Reviewers' comments:

Reviewer's Responses to Questions

**Comments to the Author**

1. Is the manuscript technically sound, and do the data support the conclusions?

Reviewer #1: Partly

2. Has the statistical analysis been performed appropriately and rigorously? 

Reviewer #1: No

3. Have the authors made all data underlying the findings in their manuscript fully available?

Reviewer #1: No

4. Is the manuscript presented in an intelligible fashion and written in standard English?

Reviewer #1: Yes

5. Review Comments to the Author

Reviewer #1: Review of the manuscript PONE-D-20-22225, entitled "Monitoring training and recovery responses with heart rate measures during standardized warm-up in elite badminton players"

The article presents a study to evaluate To investigate short-term training and recovery-related effects on heart rate during a standardized submaximal running test.

The study presents the results of original research.

The article don’t adheres to appropriate reporting guidelines and community standards for data availability as, results reported and the entire draft have been already published in other on this site:

https://www.researchgate.net/publication/343141060_Monitoring_training_and_recovery_responses_with_heart_rate_measures_during_standardized_warm-up_in_elite_badminton_players

Experiments, statistics, and other analyses are performed to a high technical standard and are described in sufficient detail. Conclusions are presented in an appropriate fashion. Nevertheless, not all are supported by the data.

The article is presented in an intelligible fashion and is written in standard English

In my opinion, specific issues must to be amended and considered before acceptance.

Comment #1:

The authors state to investigates ten elite badminton players, 7 females and 3 male. As is well known, how the serum concentration of creatine kinase and urea can change depending on the gender (in a example, previous studies reported that the females presented with a higher CPK peak and a greater relative increase in serum CPK levels after 50 maximal eccentric contractions of the arm flexor muscles, despite significantly lower baseline levels compared to males). Is missed to present the results without including males in the presented means.

Comment #2: Moreover, is missed the race difference, if there are some, as higher levels of total serum creatine kinase activity have been reported in black compared with white.

Comment #3

In the other hand, other anthropometrics measurements are missed in the study to better understanding the results (the level of muscle mass and the lower limbs length as they can change the muscle efficiency to develop CMJ or the shuttle run test, and , therefore , can change the muscle damage and the changes in CPK after exercise or the aforementioned tests).

Comment #4 Line 264

The authors state “including 4–14 measurements per player”, Were not the same measurements for all players?

Comment #5 Line 302.

The authors state “Our findings are consistent with previous studies that reported increased levels of CPK 12 hours after badminton-specific training” nevertheless this study was performed on six national male badminton players, who can increase much more the CPK levels in comparison to females.

Comment #6 Line 303-304

The authors state “As well as in response to repeated weekly microcycles in endurance athletes [10] and badminton players [11]”, developed in non-comparable 14 elite junior swimmers and triathletes and 17 elite male badminton players.

Comment #6 Line 475-478

The authors state “since the HRex response is known to be influenced by many factors, practitioners should consider the potentially overlapping effects of acute and short-term training load changes, long-term training adaptations, and external confounding factors when interpreting HRex.” I agree, and this is the reason why I think that many of the conclusions can’t be stated for this study.

6. PLOS authors have the option to publish the peer review history of their article (what does this mean?). If published, this will include your full peer review and any attached files.

Reviewer #1: **Yes: **PhD. Monica Solana-Tramunt

- - - - -

---

## [Author Response · Author response to Decision Letter 0]

16 Sep 2020

Comments to the editor

Editor:

Dear Dr. Schneider,

Thank you for submitting your manuscript to PLOS ONE. After careful consideration, we have decided that your manuscript does not meet our criteria for publication and must therefore be rejected. As you will see, some limits have been raised during the reviewing process. However, the major problem is that the entire text, results, figures and tables can be search open access in ResearchGate and SportRχiv, without peer-review and this migth be confonding for PLOS ONE.

I am sorry that we cannot be more positive on this occasion, but hope that you appreciate the reasons for this decision.

Yours sincerely,

Laurent Mourot

Academic Editor

PLOS ONE

->Authors’ response: 

Dear Dr. Mourot

Thank you for the quick editorial process and decision. 

We believe that the decision to reject the manuscript for the reasons mentioned in the decision letter was a misunderstanding, as the main reason for rejection it was stated that “the major problem is that the entire text, results, figures and tables can be search open access in ResearchGate and SportRχiv, without peer-review and this migth be confonding for PLOS ONE”. This is in contrast to PLOS submission guidelines and policies. PLOS actively encourages authors to publish preprints prior to or upon submission. We believe that this issue has already been resolved in the process of our appeal request. 

We also believe that the concerns raised by the reviewer have no substantial impact on the study results or conclusions. We provide detailed responses to the reviewer’s comments and have tried to make appropriate revision (in red).

Please find more detailed comments in the following.

a.) Preprint Publication:

We purposefully decided to publish the preprint and its underlying data in accordance to and because of the PLOS submission guidelines. 

As stated in the submission guidelines, PLOS and its associated journals “encourages authors to post preprints as a way to accelerate the dissemination of research” (for reference: https://journals.plos.org/plosone/s/preprints). This includes the publication of the full manuscript in the pre-submission version, and all its accompanying results, material, and data. The open and findable dissemination of the manuscript and supporting material on repositories like SportRxiv or ResearchGate prior to submission and peer-review is an essential component of preprints. Based these PLOS guidelines we decided to publish a preprint to adhere to open science practices.

In addition, PLOS also “require[s] authors to make all data necessary to replicate their study’s findings publicly available without restriction” (https://journals.plos.org/plosone/s/data-availability). This was also done in accordance to PLOS data sharing policy and using the Open Science Framework as one of the PLOS recommended repositories (https://journals.plos.org/plosone/s/recommended-repositories).

b.) Reviewer comments:

We appreciate the critical comments by the reviewer. Dr. Solana-Tramunt raises some valid criticism. We believe that the concerned raised by the reviewer are in total of small magnitude and can be addressed with minor revision. 

One main criticism of the reviewer was the non-availability of data (Question 2) and the inappropriate reporting guidelines and community standards by publishing (5. Comments to the Authors). Dr. Solana-Tramunt seems to have missed that the main dataset was uploaded as “Supporting Information” file and the full data and analysis documentation was deposited on the Open Science Framework (https://doi.org/10.17605/OSF.IO/UP4HT) and referenced within the submission process and within the manuscript. There might have been technical issues, so that she was not able to access the documents and information during the review process.

Content wise, the reviewer seemed to have missed an important analytical aspect: 

The descriptive statistics for creatine kinase are indeed biased by pooling genders. As suggested by the reviewer, we therefore included descriptive statistics for creatine kinase (CK) und Urea for females only (Revisions in Table 1, page 11). However, the main analysis were within-subject comparisons with standardized mean differences, which reduces potential between subject effects in mean CK and Urea levels (e.g. due to gender, race). Thus, the described training strain-related effects in the surrogate measures creatine kinase and the subjective recovery and stress ratings were clearly and appropriately verified in every single subject. The criticism raised by Dr. Solana-Tramunt indeed concerns the descriptive statistics for creatine kinase, but it has little effect on the main results and no effect on the conclusions of our study for the analytical reasons mentioned above.

In this rebuttal letter we provided detailed responses to the reviewer’s comments and we believe that we have addressed the criticisms raised by the reviewer appropriately, while none of the criticism invalidates our methods or the main inferential results.

Kind regards,

Christoph Schneider (on behalf of the coauthors)

 

Comments to the reviewer

Dear Dr. Solana-Tramunt, 

Thank you for providing a critical and fast review of our manuscript. We believe that respective revisions (in red) help to improve the article. Please find our point-by-point response to your comments below.

We tried to respond appropriately to all concerns raised. Please also see the revisions made in the manuscript.

Kind regards,

Christoph Schneider

Question 1

Is the manuscript technically sound, and do the data support the conclusions?

• Reviewer #1: Partly

->Authors’ response to question 1: 

We will reply to all comments and criticism raised by the reviewer below. 

Question 2

Has the statistical analysis been performed appropriately and rigorously? 

• Reviewer #1: No

->Authors’ response to question 2: 

In your comments to section “5. Review - Comments to the Author” below you write:

“Experiments, statistics, and other analyses are performed to a high technical standard and are described in sufficient detail.”

We therefore believe that the “negative” response here might be a mistake. Nevertheless, we will briefly describe the reasoning of our analysis to highlight why we believe that our analyses were appropriate to answer the research question. 

We aimed to model the main variable (exercise heart rate) with a within-subject linear mixed model that was built to address the main research question of interest, i.e., determine the overall mean short-term response in exercise heart rate, while accounting for potential long-term effects, as well as intra-individual and inter-individual variability (Goltz et al., 2018, Senn et al., 2011).

The secondary descriptive analyses were performed in accordance with the study design by performing intra-individual comparisons between repeated measurements at recovered versus strained states in a similar fashion as recommended for single-subject research designs (Barker et al., 2011). Therefore, the descriptive analyses served to verify intra-individual training strain responses within the given study sample. Training stain responses of at least moderate individual effect sizes were observed in every athlete for a minimum of four different monitoring measures (i.e., creatine kinase and perceived recovery and stress ratings; for reference please see the Supporting Information file “S1 Table. Summary of individual monitoring data”). Then, individual effects were summarized for descriptive purposes. On average, moderate to large individual effect sizes were observed for creatine kinase and perceived recovery and stress ratings (Table 1, Figure 3). These findings support the appropriateness of the implemented study design for describing different levels of recovery and training strain during repeated microcycles in every athlete. As substantial training strain related responses in creatine kinase and perceived recovery and stress ratings were observed in every athlete, we are confident that the Mondays and Fridays measurements actually represented contrasting states for every athlete in our study within the observational period which supports internal validity of the study design.

References: 

-Barker J, McCarthy P, Jones M, Moran A. Single case research methods in sport and exercise psychology. Abingdon England, New York: Routledge; 2011.

-Goltz FR, Thackray AE, King JA, Dorling JL, Atkinson G, Stensel DJ. Interindividual Responses of Appetite to Acute Exercise: A Replicated Crossover Study. Med Sci Sports Exerc. 2018; 50: 758–768. https://doi.org/10.1249/MSS.0000000000001504

-Senn S, Rolfe K, Julious SA. Investigating variability in patient response to treatment--a case study from a replicate cross-over study. Stat Methods Med Res. 2011; 20: 657–666. https://doi.org/10.1177/0962280210379174

Question 3

Have the authors made all data underlying the findings in their manuscript fully available?

• Reviewer #1: No

->Authors’ response to questions 3:

Maye there had been technical issues, so that the reviewer was not able to access the full material that was uploaded with the manuscript. I have recently double checked our submission, and everything was properly uploaded. Please contact the Academic Editor or the editorial office, if you do not have access to all material and links as uploaded and listed below: 

The main dataset was uploaded as Supporting Information file “S1 Dataset. Original data” (S1_dataset.xlsx) during submission.

In addition, and as referenced in the data availability statement, supplementary material availability statement (lines 653-657) and in the main text (lines 244-245), a full data and analysis documentation was deposited at the Open Science Framework at https://doi.org/10.17605/osf.io/up4ht. Therefore, each single data and analysis file from our study was made available upon submission, which allows full reproducibility of all analyses, results, tables and figures (exceptions: “Fig 1. study design” and “Fig 2. Submaximal shuttle-run test”).

Question 4

Is the manuscript presented in an intelligible fashion and written in standard English?

• Reviewer #1: Yes

->Authors’ response to question 4: No comment

Question 5

Review Comments to the Author

Reviewer #1: Review of the manuscript PONE-D-20-22225, entitled "Monitoring training and recovery responses with heart rate measures during standardized warm-up in elite badminton players":

The article presents a study to evaluate To investigate short-term training and recovery-related effects on heart rate during a standardized submaximal running test.

The study presents the results of original research.

• Reviewer #1 comment: 

The article don’t adheres to appropriate reporting guidelines and community standards for data availability as, results reported and the entire draft have been already published in other on this site:

https://www.researchgate.net/publication/343141060_Monitoring_training_and_recovery_responses_with_heart_rate_measures_during_standardized_warm-up_in_elite_badminton_players

->Authors’ response:

We purposefully decided to publish the preprint and its underlying data specifically in accordance to and because of the PLOS submission guidelines and policies. 

As stated in the submission guidelines, PLOS and its associated journals clearly “encourages authors to post preprints as a way to accelerate the dissemination of research” (for reference: https://journals.plos.org/plosone/s/preprints). This includes the publication of the full manuscript in the pre-submission version, and all its accompanying results, material, and data. The open and findable dissemination of the manuscript and supporting material on repositories like SportRxiv or ResearchGate prior to submission and prior to peer-review is an essential component of preprints and open science. Based these PLOS guidelines we decided to publish a preprint to adhere to open science practices.

In addition, PLOS also “require[s] authors to make all data necessary to replicate their study’s findings publicly available without restriction” (https://journals.plos.org/plosone/s/data-availability). As mentioned above, this was also done in accordance to PLOS data sharing policy and using the Open Science Framework as one of the PLOS recommended repositories (https://journals.plos.org/plosone/s/recommended-repositories).

Finally, we followed the PLOS statistical reporting guidelines as outlined by Lang and Altman (2013).

Reference:

Lang, T. A., & Altman, D. G. (2013). Basic statistical reporting for articles published in clinical medical journals: The SAMPL Guidelines. In P. Smart, H. Maisonneuve, & A. Polderman (Eds.), Science editors' handbook (2nd ed., pp. 175–182). EASE (European Association of Science Editors).

• Reviewer #1 comment: 

Experiments, statistics, and other analyses are performed to a high technical standard and are described in sufficient detail.

Conclusions are presented in an appropriate fashion. Nevertheless, not all are supported by the data.

->Authors’ response:

We will try to address all concerns raised by reviewer with respective responses to the comments listed below.

• Reviewer #1: 

The article is presented in an intelligible fashion and is written in standard English

In my opinion, specific issues must to be amended and considered before acceptance.

• Reviewer #1, comment #1:

The authors state to investigates ten elite badminton players, 7 females and 3 male. As is well known, how the serum concentration of creatine kinase and urea can change depending on the gender (in a example, previous studies reported that the females presented with a higher CPK peak and a greater relative increase in serum CPK levels after 50 maximal eccentric contractions of the arm flexor muscles, despite significantly lower baseline levels compared to males). Is missed to present the results without including males in the presented means.

->Authors’ response to comment #1:

Thank you for highlighting this concern. 

We agree and we are aware that males typically have higher baseline values of creatine kinase and urea. This aspect was considered when calculating individualized reference ranges for creatine kinase and urea as part of the Supplementary Information material “S1 Appendix. Individual Player Reports”. Here, gender specific prior distributions were used to perform the individualization procedure for calculating individual reference ranges. We also agree that responses to exercise may differ between gender.

We had previously avoided gender specific sub-group analyses or presentation of separate results for gender for the following reasons:

1. The concern raised by the reviewer of gender-related effects are related to the blood-borne measures creatine kinase and urea. Our main analysis for the non-heart rate variables was the calculation of intra-individual effect sizes for the mean difference between recovered and strained states. Hence, we were interested if substantial differences between measurements at the two contrasting time points were observable at the intra-individual level and for every player. 

When comparing the intra-individual effect sizes, there was no gender-related effect visible which indicates that neither females nor males tends to show higher/lower responses. Please see the Figures A and B on the next page (rebuttal letter, page 8) on why we have no reasons to believe that the effects of interest were affected by gender in our study sample.

For reasons of clarity, we therefore initially decided to only present pooled analyses and summary statistics. Nonheart rate related variables were only used to describe training strain related short-term effects, and therefore to describe the appropriateness of our study design to compare contrasting time points during repeated microcycles in our study sample. 

The separate presentation of summary statistics split by gender would go to substantial costs of clarity, while not adding any substantive information in the specific case of our study sample.

2. As outlined in 1, intra-individual effects were not affected by gender. Yet, it was very important for us to be able to provide the dataset used for this study to allow full reproducibility of all our analyses and results. The main difficulty of publishing original data from elite athletes at national level, is that it is very difficult the successfully de-identify the original dataset and make sure that athletes are not identifiable in retrospect. 

To ensure this, we avoided to publish any data that would make a potential identification of athletes more likely without being relevant for the main analyses. As described in 1., gender had no clear effect on the intra-individual effect sizes. In our study, only intra-individual effects were of interest (i.e.: Was there a substantial mean difference present between recovered and strained measurements for every athlete?). Neither the players individual mean values at recovered and strained state were of interest, nor the average mean values. 

This why we did not provide further details on the individual players age, height, weight, gender and why we also did not provide any information on the time frame or the year the study was conducted. Any of this information would make it far more likely to be able to identify at least some of the athletes in retrospect. As mentioned above, the results and analysis as currently performed are not substantially affected by gender, and we would therefore like to not include information like gender or respective sub-group analyses in the manuscript or its accompanying material for reasons of ensuring the athletes privacy. 

3. Sample size was too small to perform proper, formal sub-group analyses. Even excluding the three male players would reduce the respective sample size by 30%, which increases further uncertainty in the calculated statistics.

4. We had no reason to believe that responses in exercise heart rate would be clearly affected by gender. As exercise heart rate was the main variable of interest, we decided to only perform pooled analyses to keep analyses consistent for all variables.

 

[Please find the figures A and B in the attachment] 

--Fig A: Distribution and histograms of individual effect sizes for comparison between recovered and strained state. Variables: creatine kinase (natural logarithm), urea (natural logarithm), perceived recovery and stress ratings (PPC, OR, MS, OS), Countermovement jump, jump efficiency (multiple rebound jump), exercise heart rate, rating of perceived exertion. Red: female (f), green: male (m).

--Fig B: Individual effect sizes for comparison between recovered and strained state. Variables: creatine kinase (natural logarithm), urea (natural logarithm), perceived recovery and stress ratings (PPC, OR, MS, OS), Countermovement jump, jump efficiency (multiple rebound jump), exercise heart rate, rating of perceived exertion. Red: female (f), green: male (m).

5. Additional gender-specific analyses would, in theory, mainly be relevant for blood-borne measures, which were not of primary interest. Hence, performing additional sub-group analyses and providing respective tables would make the results and appendix even more extensive, without being directly relevant to the main findings.

To summarize: 

As mentioned in the beginning of this response to comment #1, we fully agree that gender can in general be expected to affect average values and potentially exercise responses in creatine kinase and urea. 

Nevertheless, there were no gender-related differences visible in any of the monitoring markers in our specific sample. For reasons of clarity, we therefore prefer to only present results of individual effect sizes on pooled data. We believe that additional sub-group analyses would further complicate the presentation of the results, while there is no additional information gained in our specific case. Moreover, the small sample size would not allow formal analyses of gender differences.

Revisions:

As a compromise between generally valid criticism of the reviewer (gender-related effects are typically relevant) and the circumstance of our specific study (no gender effects observable; substantially reduced sample size when only analyzing females; issues with maintaining de-identification), we added additional summary statistics for females only for creatine kinase and urea within the Table 1. Please see the revisions in Table 1, page 11: Mean ± SD for females only: CK Rec 120 ± 44 U/L, CK Strain 212 ± 92 U/L, Urea – Rec 26 ± 7 mg/dL, Strain 28 ± 6 mg/dL.

• Reviewer #1, comment #2: 

Moreover, is missed the race difference, if there are some, as higher levels of total serum creatine kinase activity have been reported in black compared with white.

->Authors’ response to comment #2:

Thank you for highlighting this. However, there were no black athletes in our sample.

• Reviewer #1, comment #3:

In the other hand, other anthropometrics measurements are missed in the study to better understanding the results (the level of muscle mass and the lower limbs length as they can change the muscle efficiency to develop CMJ or the shuttle run test, and , therefore , can change the muscle damage and the changes in CPK after exercise or the aforementioned tests).

->Authors’ response to comment #3: 

Thank you for highlighting this aspect. We agree that additional measures could be potentially useful for better understanding the results.

Due to the observational character of the study we were not able not perform any additional measurements that allowed us to estimate, for example, players’ muscle mass or lower limb length. We were only able to collect data directly related for the evaluation of monitoring markers within the given study design. We therefore cannot provide data related to muscle mass or lower limb length.

In addition, for the reasons mentioned above to ensure that data will remain as de-identifiable as possible, we purposefully did not include additional information that was not directly related to the main analysis of exercise heart rate.

Revision:

We included a respective statement to in the “Limitations and strengths of the study” section (lines 469-471): “Finally, it might also be valuable to evaluate the effects of potential influencing factors (e.g. anthropometrics, body composition) to better understand inter-individual differences in training and recovery responses.”

• Reviewer #1, comment #4 Line 264:

The authors state “including 4–14 measurements per player”, Were not the same measurements for all players?

->Authors’ response to comment #4:

The reviewer is correct. Due to the reasons mentioned in the limitations section (lines 429-436), we were not able to collect the maximum amount of data for every player. These circumstances are common and unfortunately unavoidable in longitudinal field studies with elite athletes.

As discussed in lines 429-436, this clearly represents a limitation of the study. We did our best to address this limitation by respective analytical decisions and methods:

Sub-optimal data structure will unavoidably influence inferential statistics and results while probably increasing uncertainty in the estimates. We therefore performed linear mixed models which, in contrast to “classical repeated measures analysis”, generally allow data to be used for analysis even in presence of missing data and they “naturally handle uneven spacing of repeated measurements” (Seltman, 2018), as was the case in our study.

In addition, the within-subject standard deviations, which were used for calculating individual effect sizes, were calculated by polling intra-individual variability by accounting for different number of observations per athlete. In the case of exercise heart rate, this simple estimate of within-subject variability for recovered measurements was quite similar to the within-subject variability estimate from the linear mixed model. In total, 95 exercise heart rate measurements were included in the linear mixed model (line 264). 

Finally, individual exercise heart rate time series (Figure 4) as well as full individual player reports (“S1 Appendix. Individual Player Reports”) were provided for every player to transparently allow the reader to visually evaluate the effects summarized by the inferential statistics. 

Reference:

-Chapter 15 Mixed models: Seltman, H. J. (2018). Experimental Design and Analysis. http://www.stat.cmu.edu/~hseltman/309/Book/Book.pdf

• Reviewer #1, comment #5 Line 302:

The authors state “Our findings are consistent with previous studies that reported increased levels of CPK 12 hours after badminton-specific training” nevertheless this study was performed on six national male badminton players, who can increase much more the CPK levels in comparison to females.

->Authors’ response to comment #5:

As mentioned above, we agree with the reviewer that males typically show higher average creatine kinase values and that responses may differ between females and males. 

However, we think that our statement in lines 301-302 is neither in conflict with this fact, nor that it is thereby incorrect. Majumdar et al. (1997) found increased levels of creatine kinase following badminton-specific training. In our study, we also observed increased levels of creatine kinase following mainly badminton-specific training. Although there are typically gender-related differences expectable, both studies reported increased levels of creatine kinase following badminton training and can therefore be considered consistent in general terms: Badminton-specific training may elicit higher creatine kinase values post exercise. Increased levels of creatine kinase following badminton-specific training was observed both in males [acute (Majumdar et al., 1997) and short-term (Barth et al., 2019)] and in females [short-term (our findings)]. Therefore, we consider our creatine kinase results consistent with the referenced articles, despite the differences in gender.

In contrast to the reviewer’s concern, we believe that our statement is especially adequate because there were differences in gender between studies. When males typically show higher creatine kinase values (and responses), one would not necessarily expect similar effects in females. However, as we observed changes in similar direction, despite studying mainly female athletes, the responses observed in our study are indeed consistent with the observations in males. Please note that with our analytical approach, the magnitude of response was compared to the variability in measurements. Therefore, even small but consistent responses would be identified as “clear” response. 

-Barth, V., Käsbauer, H., Ferrauti, A., Kellmann, M., Pfeiffer, M., Hecksteden, A., & Meyer, T. (2019). Individualized Monitoring of Muscle Recovery in Elite Badminton. Frontiers in Physiology, 10, 778. https://doi.org/10.3389/fphys.2019.00778

-Majumdar, P., Khanna, G. L., Malik, V., Sachdeva, S., Arif, M., & Mandal, M. (1997). Physiological analysis to quantify training load in badminton. British Journal of Sports Medicine, 31(4), 342–345. https://doi.org/10.1136/bjsm.31.4.342

• Reviewer #1, comment #6 Line 303-304:

The authors state “As well as in response to repeated weekly microcycles in endurance athletes [10] and badminton players [11]”, developed in non-comparable 14 elite junior swimmers and triathletes and 17 elite male badminton players.

->Authors’ comment to comment #6 303-304:

We again agree that studied subjects were different between the referenced studies and our study.

In the following, we will briefly outline why we consider these references adequate for the respective statement:

In our study, we compared two contrasting time points during repeated preparatory training cycles. This study design required that the two time points represent contrasting states (i.e., ‘recovered’ versus ‘strained’ state). This important aspect was communicated with the coaches prior to the study period, and only measurements were included that satisfied the requirement of recovered or strained time points based on the training schedule. For example, measurements were not considered for analyses if training or competition schedules conflicted with the inclusion criteria (please see lines 431-434). More importantly, to verify that the two time points represent contrasting states, the different monitoring markers were required to display respective short-term effects.

To make sure that we would be potentially able to observe differences in the various monitoring markers in the ‘recovered’ and ‘strained’ time points, we used a study design that has previously been shown to display different levels of creatine kinase and perceived recovery and stress ratings (lines 421-424). As the reviewer correctly mentioned, the athletes in the two previous studies that have implemented the same study design were different compared to our sample (Barth et al., 2019; Hecksteden et al., 2017). Therefore, we could not be entirely sure that their findings would translate to our study sample and our training context. However, our findings were consistent with those from the referenced two studies, as increased levels of creatine kinase were observed in all three studies, despite differences in the subjects and differences in sport when compared to the Hecksteden et al. (2017) study. We therefore believe that our statement “Our findings are consistent with previous studies …” is therefore appropriate.

References:

-Barth V, Käsbauer H, Ferrauti A, Kellmann M, Pfeiffer M, Hecksteden A, et al. Individualized Monitoring of Muscle Recovery in Elite Badminton. Front Physiol. 2019; 10: 778. https://doi.org/10.3389/fphys.2019.00778

-Hecksteden A, Pitsch W, Julian R, Pfeiffer M, Kellmann M, Ferrauti A, et al. A new method to individualize monitoring of muscle recovery in athletes. Int J Sports Physiol Perform. 2017; 12: 1137–1142. https://doi.org/10.1123/ijspp.2016-0120

• Reviewer #1, comment #6 Line 475-478:

The authors state “since the HRex response is known to be influenced by many factors, practitioners should consider the potentially overlapping effects of acute and short-term training load changes, long-term training adaptations, and external confounding factors when interpreting HRex.” I agree, and this is the reason why I think that many of the conclusions can’t be stated for this study.

->Authors’ comment to comment #6 475-478:

Please also see our responses to your specific comment above. 

As outlined in the statement in lines 475-478, we believe that heart rate measure in sports practice can be potentially influenced by many factors. Therefore, we included this statement in the conclusions, which should, from our point of view, be considered when trying to generalize our findings beyond our specific study model. This concluding statement was aimed to caution against too optimistic generalization from our specific study findings. Nevertheless, as outlined in the following, in our study we aimed to account for as many potential influencing factors as possible with several methodological and analytical decisions. Furthermore, we aimed to frame and discuss our findings in a reserved manner to avoid overstatements of our findings.

Tests were always performed at the same time of the day for all players, which eliminates potential effects due to circadian rhythm (please see lines 114-115 and Fig 1.). Furthermore, temperature was recorded for every session. Observed intra-individual difference between recovered and strained tome points were deemed neglectable compared effects of heat and cold in the previous literature (see lines 385-389). Since tests were performed in the morning sessions as the first part of the (pre-) warm-up routine, immediate effects of previous exercise (like immediate or acute exercise-related fatigue) could also be excluded.

Based on the repeated measures design and our statistical analyses, we aimed to determine short-term effects while controlling for (linear) long-term changes over the study period (lines 210-213 and 219-221). We further accounted for inter-individual differences in HRex or inter-individual differences in short-term or long-term effects by specifying respective random effects. We therefore believe that a substantial number of important (potential) influencing factors were controlled or sufficiently accounted for with our study design and statistical analysis. 

Question 6

Do you want your identity to be public for this peer review?

• Reviewer #1: Yes: PhD. Monica Solana-Tramunt

->Authors’ response to question 6:

Thank you for supporting open peer review. We believe that open peer review is very important step forward towards a modern and open science culture.

---

## [Decision Letter · Decision Letter 1]

9 Nov 2020

PONE-D-20-22225R1

Monitoring training and recovery responses with heart rate measures during standardized warm-up in elite badminton players

PLOS ONE

Dear Dr. Schneider,

Thank you for submitting your manuscript to PLOS ONE. After careful consideration, we feel that it has merit but does not fully meet PLOS ONE’s publication criteria as it currently stands. Therefore, we invite you to submit a revised version of the manuscript that addresses the points raised during the review process. Strong limits have been underlined by reviewers, especially regarding the limited amont of data and, more important, a proper description of the training load. Unless additional data are provided, it is not possible to accept this manuscript for publication.

We look forward to receiving your revised manuscript.

Kind regards,

Laurent Mourot

Academic Editor

PLOS ONE

Journal Requirements:

1. PLOS ONE style templates can be found at https://journals.plos.org/plosone/s/file?id=wjVg/PLOSOne_formatting_sample_main_body.pdf and

Reviewers' comments:

Reviewer's Responses to Questions

**Comments to the Author**

1. If the authors have adequately addressed your comments raised in a previous round of review and you feel that this manuscript is now acceptable for publication, you may indicate that here to bypass the “Comments to the Author” section, enter your conflict of interest statement in the “Confidential to Editor” section, and submit your "Accept" recommendation.

Reviewer #2: (No Response)

Reviewer #3: (No Response)

2. Is the manuscript technically sound, and do the data support the conclusions?

Reviewer #2: Yes

Reviewer #3: No

3. Has the statistical analysis been performed appropriately and rigorously? 

Reviewer #2: Yes

Reviewer #3: Yes

4. Have the authors made all data underlying the findings in their manuscript fully available?

Reviewer #2: Yes

Reviewer #3: Yes

5. Is the manuscript presented in an intelligible fashion and written in standard English?

Reviewer #2: Yes

Reviewer #3: Yes

6. Review Comments to the Author

Reviewer #2: I appreciate the invitation and the challenge.

The theme is relevant and a key factor for success in adapting to training, controlling the individual training load, and the response to training loads.

I think the authors could have explored Heart Rate Variability as a tool for assessing the autonomic nervous system, increasing the robustness of the results.

The authors assume that athletes would respond in the same way to the increase and decrease the training load. However, inter-individual variability should have been considered. Bringing uncertainty of the results achieved

Responses to reviewers improved and clarified the article, raising the findings, methodology, and discussion.

It would be more robust to have a larger amount of data, to reinforce the findings.

Thank you.

Reviewer #3: Title. Monitoring training and recovery responses with heart rate measures during standardized warm-up in elite badminton players.

Question 1. Is the manuscript technically sound, and do the data support the conclusions? The manuscript must describe a technically sound piece of scientific research with data that supports the conclusions. Experiments must have been conducted rigorously, with appropriate controls, replication, and sample sizes. The conclusions must be drawn appropriately based on the data presented.

- No.

After having carefully read the manuscript, and despite a very interesting scientific issue, I have to report some significant limitations in the document.

In my opinion, a major limitation of the manuscript is that no information is given concerning the training plan and the context in which the data were collected. The purpose of the study was to investigate short-term training and recovery related effects on heart rate during a standardized submaximal test.

One might expect having strict monitoring of training load (short term variations at an intra-individual stage, for example volume, intensity, subjective monitoring, the physiological intensity of the training sessions…). This would have allowed making interpretations and to read the heart rate markers variations at the light of this training load monitoring. This would also have allowed enriching the analyses by linear or nonlinear regression equation models to investigate causal statistical links.

Without training load monitoring, I think that it is not possible, for the reader and the scientific community, to interpret and to be fully convinced by what the authors said about their results. For example, the authors said that (L.54) “HRex decreased on average in response to intensified training during preparatory training micro-cycle”. However, I failed to find data in the manuscript that may help the reader to judge for himself (e.g., based on accurate training load monitoring and by confronting these training load markers to cardiac ones for example). In the same line, authors said that (L.304) “self-reported measures of recovery and stress were reduced and increased in response to training strain, respectively”; or that (L.305) “The main finding of this study was that HRex was sensitive to short-term changes in training load within the current training regime”; or that (L.368) “In summary, we conclude that HRex can also decrease with increased training load under normal training conditions, even if the focus is not solely on aerobic-type exercise”; or (L.410) “In summary, our findings suggest that, on average, HRex was clearly affected by short-term training load changes”. I am confused with this type of sentence because I am unable to understand what, in the analyses or in the factual descriptive elements of the training load, would allow the reader to judge the validity of this type of statement.

In this line, I think it would be more appropriate to delete all the sentences which refer to the training load (or to incorporate clear and factual data and analyses of the training load variations at an intra-individual level). On the other hand, I suggest focusing the discussion and interpretations on how HRex change around the protocol considering the other markers (e.g., CK, Urea, subjective measures), and discussing why HRex may be of interest. I sincerely think that this strategy would give more strength to the manuscript and allow to preserve the short-term aspects in the discussion.

Another limit is that the results are clear for some athletes (but not for all), and for some time points (but not for all). In this line, authors said that (L.397), “the intra-individual responses of some athletes were surprisingly clear and consistent given the uncontrolled training setting of the current study, in which standardized training stimuli were not intended and gapless data were rare”. I think that this sentence sums up the issues of the readers when confronted with the conclusions given in the manuscript: “Our findings suggest that HRex measured during a standardized warm-up is sensitive to short-term changes in training load, with HRex decreasing on average in response to intensified training during preparatory training microcycles. From a practical perspective, it seems advisable to determine intra-individual recovery–strain responses by repeated testing, as HRex responses may vary substantially between and within players”. Indeed, it is not possible to check the changes in training load, even less to check the intensification of training load. Complementary, by carefully reading athletes’ patterns and responses (Figure 4), we can quickly observed that there is a lot of situations in which HRex in strain condition and in recovered conditions were not very different or reversed based on the literature (Payer B 40-45, 50-60; Player F 10, 60; Player G : 0-10; 60-70. Etc.). Then generalization seems to be hard and I failed to understand why, based on the results of the study, it would be interesting to use HRex in short term training load monitoring and management.

Another limitation of the manuscript is that the recovery period is not monitored. The purpose of the manuscript is to investigate short-term training and recovery-related effects on HRex. I understand that weekend are a priori recovery periods, but how the reader (and the authors) can be sure that athletes don’t practice or were exposed to any stressors during these periods (which may explain some individual patterns)? In this line, the authors said that (L.119) “Monday values were categorized as ‘recovered’ state (Recovery) after 1–2 days of pronounced recovery […]”. But there are no training load markers which may help the reader to understand this categorization. Perhaps sometimes athletes were exposed to stressors during the two days the authors categorized as recovered states, and maybe the four days of training (e.g., based on aerobic training thematic) induced a physiological compensation or regeneration process? Perhaps not… I don’t know. In the same line as upper, I recommend to analyzes and to interpret HRex results at the light of the data the authors have and avoid extrapolation about the training periodization. I think that this strategy will allow the authors to consider the short-term aspects of the variations and patterns, and to discuss them regarding the other markers computed.

Authors said that they used the SRR for monitoring subjective stress and recovery states. Please, indicate examples of items to help the readers understanding the variables. Complementary, authors should give some elements about the structural validity of the tool in the study context (Cronbach for example). Finally, the fact that the authors used printed versions and then online versions need to be discussed in the limitations section of the manuscript.

Finally, I think that the introduction section needs to be more explicit about the test choice. Indeed, it exists a lot of other monitoring tests in the literature (YoYo, tilt-test…) which could be incorporated into the training sessions. I think that the incremental submaximal test was chosen for some theoretical reasons, please give them.

Question 2. Has the statistical analysis been performed appropriately and rigorously?

In SAS_Output.pdf, the dependent variable is lnHRex and not HRex. This data transformation is critical to explain in the statistical section. Moreover, tables and figures should be presented based on lnHRex and not based on HRex results.

I think that the statistical results should be included in the manuscript and not referred to additional materials. I think that it is of key importance for readers to have easy access to your mixed model effect results and to be able to check your results quickly.

I sincerely hope that my comments will be constructive for authors in their publication purchase.

Sincerely.

7. PLOS authors have the option to publish the peer review history of their article (what does this mean?). If published, this will include your full peer review and any attached files.

Reviewer #2: **Yes: **André B. Coelho

Reviewer #3: No

---

## [Author Response · Author response to Decision Letter 1]

29 Nov 2020

PONE-D-20-22225R1

Monitoring training and recovery responses with heart rate measures during standardized warm-up in elite badminton players

PLOS ONE

Document structure:

• Bold formatting indicates the comments by the editor and the reviewers.

• Changes that were made to the manuscript within this 2nd revision are highlighted in red.

• Line numbers refer to the “Revised Manuscript with Tracked Changes” document.

• Please also refer to the “Response to Reviewers” document for a better document structure and the now included Table 1 and Fig 2.

Editor:

Dear Dr. Schneider,

Thank you for submitting your manuscript to PLOS ONE. After careful consideration, we feel that it has merit but does not fully meet PLOS ONE’s publication criteria as it currently stands. Therefore, we invite you to submit a revised version of the manuscript that addresses the points raised during the review process. Strong limits have been underlined by reviewers, especially regarding the limited amount of data and, more important, a proper description of the training load. Unless additional data are provided, it is not possible to accept this manuscript for publication.

Response:

Dear Dr. Mourot,

We want to thank the editor and for their suggestions.

We have now included additional information and data regarding the training plans and training context (Table 1, Fig 2), and we updated the repository accordingly. Tables and figures were renumbered in the main text, in the table and figure captions, and in the repository documentation. 

According to the criticisms of the reviewers and their constructive feedback, we have also revised the manuscript to provide a clearer description of the study design and to focus the discussion of the results more strictly to our design. As suggested, we have also included additional information and references regarding the selection of the submaximal test protocol, regarding the structure and validation of the Short Recovery and Stress Scale, and we extended the limitations section. 

Sincerely,

Christoph Schneider (on behalf of the coauthors)

We look forward to receiving your revised manuscript.

Kind regards,

Laurent Mourot

Academic Editor

PLOS ONE

 

Reviewers' comments:

Reviewer's Responses to Questions

Comments to the Author

Question 1

1. If the authors have adequately addressed your comments raised in a previous round of review and you feel that this manuscript is now acceptable for publication, you may indicate that here to bypass the “Comments to the Author” section, enter your conflict of interest statement in the “Confidential to Editor” section, and submit your "Accept" recommendation.

• Reviewer #2: (No Response)

• Reviewer #3: (No Response)

• Authors: No response.

Question 2

2. Is the manuscript technically sound, and do the data support the conclusions?

• Reviewer #2: Yes

• Reviewer #3: No

• Authors response: We will reply to all comments and criticism raised by the reviewers below.

Question 3 

3. Has the statistical analysis been performed appropriately and rigorously? 

• Reviewer #2: Yes

• Reviewer #3: Yes

• Authors: No response.

Question 4

4. Have the authors made all data underlying the findings in their manuscript fully available?

• Reviewer #2: Yes

• Reviewer #3: Yes

• Authors: No response.

Question 5

5. Is the manuscript presented in an intelligible fashion and written in standard English?

• Reviewer #2: Yes

• Reviewer #3: Yes

• Authors: No response.

 

Question 6. Review Comments to the Author

Reviewer #2: 

I appreciate the invitation and the challenge.

The theme is relevant and a key factor for success in adapting to training, controlling the individual training load, and the response to training loads.

I think the authors could have explored Heart Rate Variability as a tool for assessing the autonomic nervous system, increasing the robustness of the results.

Response:

We thank the reviewer for their time and the constructive feedback. We attempted to respond to and address all concerns raised by the reviewer

We agree that heart rate variability (HRV) is an interesting and potentially useful tool for analyzing athletes training responses. We have previously also made positive experiences with HRV recordings in other study settings.

In this study, however, we purposefully decided against the use of HRV for the following reasons:

• Literature recommendations: In this study, we could only collect data at a maximum of two days during the week (Mondays and Fridays). At the time of investigation, resting HRV measures of less than a minimum of 3-4 recordings per week were not recommended (Buchheit, 2014). Buchheit (2014) also discouraged against the use of HRV recordings during exercise and, as suggested, post-exercise HRV recordings were a priori deemed redundant in our case due to the attempted assessment of heart rate recovery immediately after the submaximal exercise test.

• Reasons of feasibility: The implemented pre-practice monitoring routine was already at a maximum duration which was considered tolerable by the coaches. At the time of the investigation 5-min resting R-R interval recordings were considered necessary for valid HRV assessment (Buchheit, 2014). Therefore, collecting resting HRV data would have required a minimum of approximately 7 additional minutes (including 2 min of heart rate stabilization) within the pre-practice routine. This additional time was not considered feasible by the coaches in light of the sub-optimal circumstances for collecting resting or pre-exercise HRV recordings in our case.

• Technology: We used a team-based heart rate monitoring system that was not validated for deriving R-R interval data to analyze HRV.

• Home-based measurements: Additional home-based resting HRV recordings were also not considered feasible by the coaches for reasons of compliance. In addition, we only had access to Polar RS800 devices for implementing home-based HRV recordings at the time of investigation. These devices did not allow routine and time-efficient exchange of HRV recordings, which would have been a requirement from the coaches to allow timely analysis.

-Buchheit, M. (2014). Monitoring training status with HR measures: Do all roads lead to Rome? Frontiers in Physiology, 5, 73. https://doi.org/10.3389/fphys.2014.00073

The authors assume that athletes would respond in the same way to the increase and decrease the training load. However, inter-individual variability should have been considered. Bringing uncertainty of the results achieved.

Response:

We agree that in general inter-individual variability increases uncertainty in statistical estimates and therefore uncertainty in study findings. However, we want to emphasize that we addressed the possibility of inter-individual variability in responses in our analyzes:

a) In the descriptive analyzes, we summarized differences and change scores at the intra-individual level, before summarizing the average effects [lines 219-222]. 

b) In second part of the analyzes, a within-player linear mixed model was used to estimate the overall mean effect between strained and recovered states and we additionally quantified interindividual differences in this response [lines 226-230]. To account for inter-individual differences, random effects for Player ID, Player ID x Week and Player ID x State were included in our statistical model [lines 238-242]. This allowed us to quantify inter-individual variability in responses as standard deviations (in percentage, i.e., as a coefficient of variation) [lines 244-245] and we additionally estimated the proportion of true responders in HRex using a recently recommended approach [lines 256-260]. The results of the interindividual variability in effects can be found in lines 294-296, 299-300, Table 2 (as standard deviations of the standardized mean differences ‘d’), and Figs 4, 5, 7.

• Methods, lines 226-230: “In the second part of our analysis, we determined the overall mean difference in HRex between recovered and strained states and quantified interindividual differences in this response.”

• Lines 238-242: “We included random effects for Player ID (intercept) and Player ID × Week (slope) to allow for individual differences in HRex and the linearized change over the 12-week period. A random effect was also added for Player ID × State to determine interindividual variability in the mean (fixed) state effect.”

• Lines 244-245: “All random effects were specified with a variance components covariance structure and expressed as CVs (i.e., SDs in percentage).”

• Lines 256-260: “Finally, we estimated the proportion of true responders in HRex using a recently recommended approach [27–29]. This method uses the estimates of the mean short-term difference and the associated SD representing the interindividual variability to derive the proportion of interindividual differences (i.e., recovery–strain) free from (random) within-subject variability and greater than the minimum practically important difference (i.e., -1%).”

• Results, lines 294-296: “We observed a linear reduction of -1.4% (-3.0% to 0.3%) in HRex over the 12-week study period. The interindividual variability in this trend (SD as a CV) was 2.2% (-0.6% to 3.2%).”

• Lines 299-330: “The SD representing interindividual variability in this difference was 0.7% (-0.6 to 1.2; Fig 67).”

We have often focused on the average effects because researchers have previously been advised to focus on estimating average effects unless robust evidence for inter-individual differences in response is presented (Atkinson & Batterham 2015; Atkinson et al., 2019; Senn, 2018). 

-Atkinson, G., & Batterham, A. M. (2015). True and false interindividual differences in the physiological response to an intervention. Experimental Physiology, 100(6), 577–588. https://doi.org/10.1113/EP085070

-Atkinson, G., Williamson, P., & Batterham, A. M. (2019). Issues in the determination of ‘responders’ and ‘non-responders’ in physiological research. Experimental Physiology, 104(8), 1215–1225. https://doi.org/10.1113/EP087712

-Senn, S. (2018). Statistical pitfalls of personalized medicine. Nature, 563(7733), 619–621. https://doi.org/10.1038/d41586-018-07535-2

Responses to reviewers improved and clarified the article, raising the findings, methodology, and discussion.

It would be more robust to have a larger amount of data, to reinforce the findings.

Thank you.

Response:

Unfortunately, it was not possible to collect more complete or extensive datasets for various reasons. For logistic reasons, we could only monitor players from one of the two national training centers using submaximal heart rate recordings. There is only a small total number of players training at the national training centers and training at national squad level. Some players did not provide written consent for study participation or did not provide adequate number of data points to be included in the analyses. The investigation period was also limited by the duration of the preparation period prior to the World Championships. Furthermore, several recordings and measurements were missing due to variations in individual training and competition schedules (i.e., Monday or Friday tests not reflecting the recovered and strained states, respectively), diseases, injury or poor heart rate data quality (lines 458-463). 

 

Reviewer #3:

Title. Monitoring training and recovery responses with heart rate measures during standardized warm-up in elite badminton players.

Question 1. Is the manuscript technically sound, and do the data support the conclusions? The manuscript must describe a technically sound piece of scientific research with data that supports the conclusions. Experiments must have been conducted rigorously, with appropriate controls, replication, and sample sizes. The conclusions must be drawn appropriately based on the data presented.

- No.

After having carefully read the manuscript, and despite a very interesting scientific issue, I have to report some significant limitations in the document.

Response:

We thank the reviewer for their detailed assessment of the manuscript and their extensive and constructive feedback. In addition to an incomplete description of the training context, it seems that some parts of our manuscript were not written clearly enough which may cause some confusion. We attempted to respond to and address all concerns raised by the reviewer. We revised the manuscript accordingly and added additional information, data and references.

Please find our detailed responses and comments below.

In my opinion, a major limitation of the manuscript is that no information is given concerning the training plan and the context in which the data were collected. The purpose of the study was to investigate short-term training and recovery related effects on heart rate during a standardized submaximal test.

We understand this criticism and have now included additional information on the training schedules (Lines 118-124, Table 1, Fig 2) to allow the referees and the readers a better understanding of the training structure during the investigation period. In addition, we updated our repository accordingly, which now entails the underlying data and analysis files used to generate Fig 2 (https://osf.io/zfwuy/). 

The additional data should allow evaluating the appropriateness of our study design in more detail, which was aimed to compare monitoring data at two contrasting time points within repeated training week. As now detailed in Table 1, and as visualized in Fig 2, we believe that the high training volume during the week, and the maximum of one low-intensity training session on the weekend represent a clear training structural pattern which can be described as “training strain” and “recovery” period within weekly microcycles, respectively. Please find respective revisions, Table 1, and Fig 2 below:

Revisions:

• Lines 118-124: “Training was planned by the national coaches without research team interference. According to the coaches, this preparatory period was categorized as an intensified training period. Training plans were provided, and the players documented their training in the best possible way. Two representative weekly training plans are detailed in Table 1, and an exemplary 12-week time course of the training volume distribution is illustrated in Fig 2. Monday values were categorized as ‘recovered’ state (Recovery) after 1–2 days of pronounced recovery, whereas Friday values represented a ‘strained’ state (Strain) following 4 consecutive days of training with up to 2 sessions per day (Table 1, Fig 2).”

• Table 1. Two exemplary weekly training plans for player J.

[Please find the table in the "Response to Reviewers" document]

 

• Fig 2. Time course of the training volume distribution over the 12-week study period in player J.

[Please find the figure in the "Response to Reviewers" document]

One might expect having strict monitoring of training load (short term variations at an intra-individual stage, for example volume, intensity, subjective monitoring, the physiological intensity of the training sessions…). This would have allowed making interpretations and to read the heart rate markers variations at the light of this training load monitoring. This would also have allowed enriching the analyses by linear or nonlinear regression equation models to investigate causal statistical links.

Response:

We agree that training load quantification would have allowed a more detailed description of the external and internal workloads and would enable additional (statistical) analyses. We also understand and agree that direct comparisons of training load measures and training response measures are of great interest and of practical relevance. Nevertheless, the associative analysis of dose-response relationships was beyond the scope of our study. 

Given the personnel and financial resources, and given the existing infrastructure of the training facilities at the time of investigation, our observational field study was solely aimed to determine if HRex can differentiate between different states on the fatigue–recovery continuum which are typically observed within repeated training weeks. Unfortunately, complete training load data was not available for out study period. 

Revisions:

• Methods, lines 119-120: “Training plans were provided, and the players documented their training in the best possible way.“

• Limitations, lines 435: “In the absence of quantitative training load data and an objective and accurate criterion measure, …”

• Lines 441-443: “In addition, detailed training load quantification would have allowed a more complete description of the training execution and demands. Unfortunately, gapless workload monitoring data was not available for the study period.”

Without training load monitoring, I think that it is not possible, for the reader and the scientific community, to interpret and to be fully convinced by what the authors said about their results. For example, the authors said that (L.54) “HRex decreased on average in response to intensified training during preparatory training micro-cycle”. However, I failed to find data in the manuscript that may help the reader to judge for himself (e.g., based on accurate training load monitoring and by confronting these training load markers to cardiac ones for example). In the same line, authors said that (L.304) “self-reported measures of recovery and stress were reduced and increased in response to training strain, respectively”; or that (L.305) “The main finding of this study was that HRex was sensitive to short-term changes in training load within the current training regime”; or that (L.368) “In summary, we conclude that HRex can also decrease with increased training load under normal training conditions, even if the focus is not solely on aerobic-type exercise”; or (L.410) “In summary, our findings suggest that, on average, HRex was clearly affected by short-term training load changes”. I am confused with this type of sentence because I am unable to understand what, in the analyses or in the factual descriptive elements of the training load, would allow the reader to judge the validity of this type of statement.

Response:

As mentioned above, we now also included additional information about weekly training schedules and exemplary time course of training volume distribution to allow a better understanding of the training context. We believe, that the general training structure with high training volumes and mainly moderate to high qualitative target training intensities during the week, in combination with a maximum of one low-intensity training session on the weekend, allows a general and plausible statement about qualitative differences in training load for the “training week” versus “weekend” periods (Lines 118-124, Table 1, Fig 2). We also revised the manuscript to describe more clearly what is meant by, for example, “short-term changes in training load”, “training strain” or “recovery” period. Please see the revisions listed below.

We also believe that the effects observed in the response measures creatine kinase and the perceived recovery and stress ratings (SRSS: Short Recovery and Stress Scale) support and verify the assumptions of the study design to display contrasting time points within repeated training weeks, at minimum in a qualitative sense (Table 2, Fig 4).

In addition, coaches and athletes were instructed prior to the beginning of the study and were aware of the requirement that the different testing time points must represent different states of typical/habitual “training strain” or “recovery”. As described in limitations section (lines 460-462), several recordings, and measurements were not considered for analyses since this requirement was not fulfilled. We are convinced, that both players and coaches were able to qualify the validity of the inclusion criteria due to years of experience and their sports expertise. 

• Lines 460-462: “For example, several recordings were missing due to variations in individual training and competition schedules (i.e., Monday or Friday tests not reflecting the recovered and strained states, respectively), …”

In combination, we believe that these three aspects and the revisions made to the manuscript should allow evaluating the appropriateness our study design for comparing measurements at two contrasting time points within a natural preparatory training environment in elite badminton players in more detail.

Revisions:

• Abstract, lines 49-52: “Our findings suggest that HRex measured during a standardized warm-up is can be sensitive to short-term changes in accumulation of training load, with HRex decreasing on average in response to consecutive days of intensified training during within repeated preparatory training weekly microcycles.”

• Introduction, lines 94-96: “The aim of this study was to evaluate the sensitivity of HRex during a standardized submaximal shuttle-run test in response to habitual short-term changes in training load within repeated training weeks.”

• Methods, lines 118-122: “. According to the coaches, this preparatory period was categorized as an intensified training period. Training plans were provided, and the players documented their training in the best possible way. Two representative weekly training plans are detailed in Table 1, and an exemplary 12-week time course of the training volume distribution is illustrated in Fig 2.“

• Discussion, lines 313-316: “The purpose of this observational study was to assess whether HRex during standardized submaximal shuttle-runs is sensitive to short-term accumulation of training load changes within repeated habitual training weeks and whether potential responses can be consistently observed at the individual level.”

• Lines 369-371: “In comparison, the short-term HRex effect that we observed during the within repeated training microcycles …”

• Lines 380-382: “This potential sensitivity of HRex to reflect naturally occurring short-term training load changes within habitual microcycles despite a variety of training contents, not solely focusing on endurance-type exercise (Table 1, Fig 2), …”

• Lines 387-389: “In summary, we conclude that HRex can also decrease with increased training load after several consecutive training days under normal training conditions, even if the focus is not solely on aerobic-type exercise.”

• Lines 429-431: “In summary, our findings suggest that, on average, HRex was clearly affected by naturally occurring short-term training load changes within repeated training microcycles.”

In this line, I think it would be more appropriate to delete all the sentences which refer to the training load (or to incorporate clear and factual data and analyses of the training load variations at an intra-individual level). On the other hand, I suggest focusing the discussion and interpretations on how HRex change around the protocol considering the other markers (e.g., CK, Urea, subjective measures), and discussing why HRex may be of interest. I sincerely think that this strategy would give more strength to the manuscript and allow to preserve the short-term aspects in the discussion.

Response:

As described our response to the previous comment, we revised the manuscript to clarify the language regarding the use of “training load”. Please also see the revisions listed above.

Beside the overall training structural elements, we believe that the presentation and discussion of the HRex results were already dependent on and supported by the effects observed in the other markers creatine kinase and subjective measures. Please see the following paragraphs for reference:

• Discussion, Lines 316-318: “The standardized effect sizes in CK and subjective recovery–stress markers characterize the appropriateness of the present study design for describing different levels of recovery and training strain during repeated preparatory training microcycles”

• Lines 331-333: “Our study was based on the premise that training strain would be evident after four consecutive training days during each weekly microcycle. This premise was verified by substantial changes in CK and perceived recovery–stress ratings (Table 12, Fig 34).”

• Lines 446-449: “…, which was also supported by our findings for mean CK levels. In addition, moderate to large [31] standardized mean differences in perceived recovery–stress states were present in the analyzed SRSS items underpinning the appropriateness of the study design.”

Furthermore, we extended the limitations section to discuss that future studies should aim to incorporate additional quantitative load monitoring.

Revisions:

• Limitations, line 435: “In the absence of quantitative training load data and an objective and accurate criterion measure, …”

• Lines 441-444: “In addition, detailed training load quantification would have allowed a more complete description of the training execution and demands. Unfortunately, gapless workload monitoring data was not available for the study period.”

• Lines 495-500: “Nevertheless, the influence of different training characteristics (e.g., intensity, volume, and exercise mode) is still unknown, and valid quantification of concurrent different training components is very challenging, which will challenge impede the comparability and generalizability of applied field studies. Future studies should therefore also aim to systematically and comprehensively quantify training load to allow a more direct description of the training context and to enable dose-response analyses.”

Another limit is that the results are clear for some athletes (but not for all), and for some time points (but not for all). In this line, authors said that (L.397), “the intra-individual responses of some athletes were surprisingly clear and consistent given the uncontrolled training setting of the current study, in which standardized training stimuli were not intended and gapless data were rare”. I think that this sentence sums up the issues of the readers when confronted with the conclusions given in the manuscript: “Our findings suggest that HRex measured during a standardized warm-up is sensitive to short-term changes in training load, with HRex decreasing on average in response to intensified training during preparatory training microcycles. From a practical perspective, it seems advisable to determine intra-individual recovery–strain responses by repeated testing, as HRex responses may vary substantially between and within players”. Indeed, it is not possible to check the changes in training load, even less to check the intensification of training load. 

Response:

The now included training data should help to illustrate the clear weekly training structure which is the basis for the naturally occurring differences in training load when comparing the two time points. Although quantitative training load data would undoubtedly be useful in documenting the actual training loads, the effects observed in the other monitoring markers support the premise of this study design.

Based on some of the reviewer’s comments, we suspect that some formulations in our manuscript were not clear enough and that there might have been some misunderstandings. On the one hand, “intensification of training load” was referring the overall preparatory training period as classified by the coaches. We added a respective sentence to clarify this. On the other hand, intensified training load referred to the periods of training strain within a training week compared to the low training loads on the weekend. 

Revisions:

• Lines 118-119: “According to the coaches, this preparatory period was categorized as an intensified training period”). 

• Please also see the revisions listed in the previous comment above.

Finally, we want to highlight, that within this manuscript we are only referring to changes in training load within, not between training weeks. Indeed, if the aim of a study is to quantify and interpret week-to-week variations in training load and response measures, the quantification of training load data seems necessary. We revised the manuscript to clarify that we are referring only to short-term changes within, not between microcycles. Please see the following revisions for reference.

Revisions:

• Abstract, lines 50-52: “…, with HRex decreasing on average in response to consecutive days of intensified training during within repeated preparatory training weekly microcycles.”

• Introduction, line 96: “… in response to habitual short-term changes in training load within repeated training weeks.”

• Discussion, lines 314-315: “… is sensitive to short-term accumulation of training load changes within repeated habitual training weeks …”

• Lines 370-371: “…, the short-term HRex effect that we observed during the within repeated training microcycles …”

• Lines 380-381: “This potential sensitivity of HRex to reflect naturally occurring short-term training load changes within habitual microcycles …”

• Lines 387-388: “In summary, we conclude that HRex can also decrease with increased training load after several consecutive training days under normal training conditions …”

• Lines 430-431: “… by naturally occurring short-term training load changes within repeated training microcycles.”

Complementary, by carefully reading athletes’ patterns and responses (Figure 4), we can quickly observed that there is a lot of situations in which HRex in strain condition and in recovered conditions were not very different or reversed based on the literature (Payer B 40-45, 50-60; Player F 10, 60; Player G : 0-10; 60-70. Etc.). Then generalization seems to be hard and I failed to understand why, based on the results of the study, it would be interesting to use HRex in short term training load monitoring and management.

Response:

We agree that based on our results alone it is very difficult to generalize beyond our own research framework and study sample. We intentionally wanted to highlight this by presenting and visualizing individual data and change score in detail, and by presenting data on inter- and intra-individual variability in responses. Please see Table 2 (within- and between-player standard deviations), Figs 4–7, and the following references which highlight the substantial between-player and within-player variability:

• Abstract, lines 47-48: “…, and the standard deviation (as a CV) representing interindividual variability in this response was 0.7% (-0.6% to 1.2%).”

• Lines 53-54: “…, as HRex responses may vary substantially between and within players.”

• Results, lines 293-294: “The overall (grand mean) HRex was 173 bmp, with between-player and within-player SDs (as CVs) of 5.8% (90% CI: 2.7% to 7.7%) and 1.3% (1.2% to 1.5%), respectively” 

• Lines 295-296: “The interindividual variability in this trend (SD as a CV) was 2.2% (-0.6% to 3.2%).”

• Lines 299-300: “The SD representing interindividual variability in this difference was 0.7% (-0.6 to 1.2; Fig 67).”

• Lines 307-308: “… we estimated the proportion of true and substantial HRex responders to be 78%, with the remaining 22% of responses being trivial.”

• Discussion, lines 344-346: “However, it should be noted that the 90% confidence interval for the interindividual response variability was relatively wide (-0.6% to 1.2%), indicating a considerable uncertainty in this estimate.”

• Conclusions, lines 508-509: “…as HRex responses may vary substantially between and within players.” 

In addition, we have conscientiously tried avoiding overinterpretation of the results and to only draw direct conclusions in relation to our own study context and our own sample. We also avoided claims related to the monitoring of training load, or the management of training load and training stress, since all implemented monitoring markers are classified as “response” markers (i.e., players responses following training or recovery). We strongly believe that such impactful decisions or generalizations regarding training prescription or management cannot be directly supported by our data. Please see the following reference to highlight our attempt to use reserved wording and avoid overinterpretation:

• Abstract, lines 53-54: “…, as HRex responses may vary substantially between and within players.”

• Discussion, lines 344-346: “However, it should be noted that the 90% confidence interval for the interindividual response variability was relatively wide (-0.6% to 1.2%), indicating a considerable uncertainty in this estimate.”

• Lines 420-422: “Conversely, it could be argued that a mean HRex difference of -1.5% between the recovered and the strained state approximated to -2.6 bpm in our sample and may not appear very compelling.”

• Lines 422-426: “HRex is typically derived as an integer, and the observed mean difference is therefore only about twice the smallest observable difference at the individual level. At the same time, the interpretation of single HRex measurements or change scores is generally affected by an expected non-trivial measurement error [1].”

• Limitations, lines 464-465: “…, missing HRex data more strongly limit the analysis and interpretation.”

• Lines 477-479: “… , it cannot be ruled out that observed HRex responses were confounded by strain-induced changes in movement patterns or altered player behavior …”

• Conclusions, lines 509-512: “Furthermore, since the HRex response is known to be influenced by many factors, practitioners should consider the potentially overlapping effects of acute and short-term training load changes, long-term training adaptations, and external confounding factors when interpreting HRex.”

Despite various limitations, we believe that our study presents an interesting case, which documents that HRex could also be altered in response to typical variations in training strain within microcycles. As outlined in the discussion section, we are only aware of comparable HRex responses in situations with more extreme training overload (Lines 350-374, 380-387). In practice, this could be relevant for interpreting observed changes in HRex. We specifically wanted to highlight the possibility of short-term effects in combination with the substantial between-player and within-player variations observed, because this could further complicate interpretation of HRex data in practice. We therefore explicitly suggested that the repeated assessment of intra-individual responses seems necessary for valid interpretation:

• Abstract, lines 52-54: “From a practical perspective, it seems advisable to determine intra-individual recovery–strain responses by repeated testing, as HRex responses may vary substantially between and within players.”

• Discussion, lines 426-429: “To address the generic challenge of observed measurement error in sports practice, it seems advisable to establish intra-individual recovery–strain response profiles through repeated testing as part of a ‘learning phase’ [11], before decisions on training and recovery prescription are made.”

• Lines 431-433: “If decision-making at the individual level is desired, it seems advisable to incorporate repeated testing (i.e., multiple recovered versus strained measurements), as HRex responses may vary substantially between and within players.”

• Conclusions, lines 507-509: “Despite a clear average effect, we encourage practitioners to implement repeated testing when decision-making at the individual level is desired, as HRex responses may vary substantially between and within players.”

Another limitation of the manuscript is that the recovery period is not monitored. The purpose of the manuscript is to investigate short-term training and recovery-related effects on HRex. I understand that weekend are a priori recovery periods, but how the reader (and the authors) can be sure that athletes don’t practice or were exposed to any stressors during these periods (which may explain some individual patterns)? In this line, the authors said that (L.119) “Monday values were categorized as ‘recovered’ state (Recovery) after 1–2 days of pronounced recovery […]”. But there are no training load markers which may help the reader to understand this categorization. Perhaps sometimes athletes were exposed to stressors during the two days the authors categorized as recovered states, and maybe the four days of training (e.g., based on aerobic training thematic) induced a physiological compensation or regeneration process? Perhaps not… I don’t know. In the same line as upper, I recommend to analyzes and to interpret HRex results at the light of the data the authors have and avoid extrapolation about the training periodization. I think that this strategy will allow the authors to consider the short-term aspects of the variations and patterns, and to discuss them regarding the other markers computed.

Response:

We agree that workloads or additional stressors during the recovery periods were not directly monitored which can be considered a limitation. Similar to the expected training strain during the week, the assumption of “recovered” and “strained” states is based on an initial instruction of the coaches and athletes about the inclusion requirements for the two time points prior to the beginning of the study. As described earlier, several recordings, and measurements were not considered for analyses since this requirement was not fulfilled (lines 460-462). 

Furthermore, the additional monitoring markers were previously described (Heidari et al., 2019) and validated as markers of recovery status (creatine kinase: Hecksteden et al., 2017; Barth et al., 2019; SRSS: Kellmann & Kölling, 2019/2020), and provided evidence that at least a certain amount of recovery was present in most cases and in all athletes. Although we agree that we cannot provide any more direct quantification of the activities or potential stressors during the recovery periods, we do believe that the differences in the other monitoring markers are sufficient for supporting the overall validity of the study design and our claims. Please also see the revisions described above on the implementing a clearer wording.

-Barth, V., Käsbauer, H., Ferrauti, A., Kellmann, M., Pfeiffer, M., Hecksteden, A., & Meyer, T. (2019). Individualized Monitoring of Muscle Recovery in Elite Badminton. Frontiers in Physiology, 10, 778. https://doi.org/10.3389/fphys.2019.00778

-Hecksteden, A., Pitsch, W., Julian, R., Pfeiffer, M., Kellmann, M., Ferrauti, A., & Meyer, T. (2017). A new method to individualize monitoring of muscle recovery in athletes. International Journal of Sports Physiology and Performance, 12(9), 1137–1142. https://doi.org/10.1123/ijspp.2016-0120

-Heidari, J., Beckmann, J., Bertollo, M., Brink, M., Kallus, W., Robazza, C., & Kellmann, M. (2019). Multidimensional Monitoring of Recovery Status and Implications for Performance. International Journal of Sports Physiology and Performance, 14(1), 2–8. https://doi.org/10.1123/ijspp.2017-0669

-Kellmann, M., & Kölling, S. (2019). Recovery and stress in sport: A manual for testing and assessment. Routledge. https://doi.org/10.4324/9780429423857

-Kellmann, M., & Kölling, S. (2020). Das Akutmaß und die Kurzskala zur Erfassung von Erholung und Beanspruchung für Erwachsene und Kinder/Jugendliche: [The Acute Measure and the Short Scale of Recovery and Stress for Adults and Children/Adolescents] (1. Auflage). Schriftenreihe des Bundesinstituts für Sportwissenschaft.

Authors said that they used the SRR for monitoring subjective stress and recovery states. Please, indicate examples of items to help the readers understanding the variables. Complementary, authors should give some elements about the structural validity of the tool in the study context (Cronbach for example). Finally, the fact that the authors used printed versions and then online versions need to be discussed in the limitations section of the manuscript.

Response:

We thank the reviewer for pointing this out. We agree with the criticisms and revised the manuscript accordingly. 

The items mentioned in the original manuscript display the actual items of the questionnaire that were rated by the players: Physical Performance Capability (PPC), Overall Recovery (OR), Muscular Stress (MS), and Overall Stress (OS)). We now also included more detailed information on the structure of the SRSS, and included the descriptive adjectives for the selected items, as well as information the internal consistency (lines 150-160). Finally, we discussed the change in survey method in the limitations (lines 451-457).

Revision:

• Methods (lines 150-160): “Perceived recovery–stress states were assessed using the SRSS, which consists of a Short Recovery Scale and a Short Stress Scale with 8 4 items each and with responses ranging from 0 (does not apply at all) to 6 (fully applies) [12,13]. In this study, only the physical and overall recovery and stress items were analyzed, i.e., Physical Performance Capability (PPC), Overall Recovery (OR), Muscular Stress (MS), and Overall Stress (OS). Each item is provided with 4 descriptive adjectives: PPC: strong, physically capable, energetic, full of power; OR: recovered, rested, muscle relaxation, physically relaxed; MS: muscle exhaustion, muscle fatigue, muscle soreness, muscle stiffness; OS: tired, worn-out, overloaded, physically exhausted. The internal consistency for the Short Recovery Scale and the Short Stress Scale were deemed acceptable (Cronbach’s Alpha 0.72 and 0.75, respectively [13]) and previous research indicated the SRSS’s sensitive to training overload [15].“

• Limitations (lines 451-457): “It must be acknowledged, however, that we had to change the survey method from a printed to a digital version of the SRSS for reasons of compliance. This was done to enable players and coaches a more frequent implementation of the SRSS in daily practice including immediate online access to data and results. Although this change in method might have influenced the SRSS ratings, we assume the intra-individual effects to be minor in our case, as players were well familiarized with the original validated print version before using the online version.”

Finally, I think that the introduction section needs to be more explicit about the test choice. Indeed, it exists a lot of other monitoring tests in the literature (YoYo, tilt-test…) which could be incorporated into the training sessions. I think that the incremental submaximal test was chosen for some theoretical reasons, please give them.

Response:

We now added additional information about the test choice.

Revisions:

• Methods (lines 178-183): “In the absence of clear test recommendations, based on previous research experience [6], and based on pragmatic considerations by the coaches regarding the test nature, we used a tailor-made shuttle-run protocol to assess HRex and RPE (6–20 scale) as part of a standardized on-court warm-up routine (Fig 23). In contrast to established intermittent tests (e.g., sub-maximal Yo-Yo tests [17,18]), coaches and we preferred a continuous exercise test to obtain more stable heart rate data while using short-distance shuttles for movement specificity.”

Question 2. Has the statistical analysis been performed appropriately and rigorously?

In SAS_Output.pdf, the dependent variable is lnHRex and not HRex. This data transformation is critical to explain in the statistical section. Moreover, tables and figures should be presented based on lnHRex and not based on HRex results.

Response:

The information on the log-transformation of HRex prior to modeling was described in lines 232-235. Due to using lnHRex data in the mixed models for reasons of direct comparison with the minimum practically important difference of 1%, the main results and effect estimates were presented as percentages (lines 294-300, Fig 7). For descriptive purposes, log data of the marginal means, were back-transformed in HRex as bpm to allow better interpretation of raw values. As described in Altman (1999), for example, back-transformation of log-transformed data seems common practice to allow better interpretation.

For reasons of clarity, the descriptive figures 5 and 6, and descriptive statistics in Table 2 were displayed in HRex raw units, as there was no evidence indicating non-normality at the intra-individual level. Please see the DataAppendix page 2 for details: https://osf.io/3ebu7/ (the respective histogram is also directly available at: https://osf.io/tsqdh/). 

-Altman, D. G. (1999). Practical statistics for medical research. Chapman & Hall/CRC.

I think that the statistical results should be included in the manuscript and not referred to additional materials. I think that it is of key importance for readers to have easy access to your mixed model effect results and to be able to check your results quickly.

Response:

All relevant results from the mixed models are included in results section and all mixed model estimates are presented with confidence intervals (lines 293-300). The number of included observations is presented in line 281. The main short-term effects are further visualized in Fig 7. The presentation of the statistical results is in accordance with the PLOS guidelines for statistical reporting (Lang & Altman, 2013).

-Lang, T. A., & Altman, D. G. (2013). Basic statistical reporting for articles published in clinical medical journals: The SAMPL Guidelines. In P. Smart, H. Maisonneuve, & A. Polderman (Eds.), Science editors' handbook (2nd ed., pp. 175–182). EASE (European Association of Science Editors).

I sincerely hope that my comments will be constructive for authors in their publication purchase.

Sincerely.

Response:

We again want to thank the reviewer for providing a critical but constructive review of our manuscript. We truly believe that revisions helped to improve our work substantially, and that the manuscript now presents a more complete documentation of the research project.

Sincerely,

Christoph Schneider (on behalf of the coauthors)

Question 7 

7. PLOS authors have the option to publish the peer review history of their article (what does this mean?). If published, this will include your full peer review and any attached files.

Do you want your identity to be public for this peer review? For information about this choice, including consent withdrawal, please see our Privacy Policy.

• Reviewer #2: Yes: André B. Coelho

• Reviewer #3: No

• Authors: No response.

---

## [Decision Letter · Decision Letter 2]

9 Dec 2020

Monitoring training and recovery responses with heart rate measures during standardized warm-up in elite badminton players

PONE-D-20-22225R2

Dear Dr. Schneider,

We’re pleased to inform you that your manuscript has been judged scientifically suitable for publication and will be formally accepted for publication once it meets all outstanding technical requirements.

Kind regards,

Laurent Mourot

Academic Editor

PLOS ONE

Additional Editor Comments (optional):

Reviewers' comments:

Reviewer's Responses to Questions

**Comments to the Author**

1. If the authors have adequately addressed your comments raised in a previous round of review and you feel that this manuscript is now acceptable for publication, you may indicate that here to bypass the “Comments to the Author” section, enter your conflict of interest statement in the “Confidential to Editor” section, and submit your "Accept" recommendation.

Reviewer #3: All comments have been addressed

2. Is the manuscript technically sound, and do the data support the conclusions?

Reviewer #3: (No Response)

3. Has the statistical analysis been performed appropriately and rigorously? 

Reviewer #3: (No Response)

4. Have the authors made all data underlying the findings in their manuscript fully available?

Reviewer #3: (No Response)

5. Is the manuscript presented in an intelligible fashion and written in standard English?

Reviewer #3: (No Response)

6. Review Comments to the Author

Reviewer #3: (No Response)

7. PLOS authors have the option to publish the peer review history of their article (what does this mean?). If published, this will include your full peer review and any attached files.

Reviewer #3: **Yes: **Philippe Vacher

---

## [Editor Report · Acceptance letter]

11 Dec 2020

PONE-D-20-22225R2 

Monitoring training and recovery responses with heart rate measures during standardized warm-up in elite badminton players 

Dear Dr. Schneider:

I'm pleased to inform you that your manuscript has been deemed suitable for publication in PLOS ONE. Congratulations! Your manuscript is now with our production department. 

Kind regards, 

on behalf of

Dr Laurent Mourot 

Academic Editor

PLOS ONE